# LETTERS

# Stem cell-derived synthetic embryos self-assemble by exploiting cadherin codes and cortical tension

Min Bao[1,2], Jake Cornwall-Scoones[1,3], Estefania Sanchez-Vasquez[1], Andy L. Cox[1], Dong-Yuan Chen[1], Joachim De Jonghe[3,4], Shahriar Shadkhoo[1], Florian Hollfelder [4], Matt Thomson[1], David M. Glover[1] and Magdalena Zernicka-Goetz [1,2] ✉

Mammalian embryos sequentially differentiate into trophectoderm and an inner cell mass, the latter of which differentiates into primitive endoderm and epiblast. Trophoblast stem (TS), extraembryonic endoderm (XEN) and embryonic stem (ES) cells derived from these three lineages can self-assemble into synthetic embryos, but the mechanisms remain unknown. Here, we show that a stem cell-specific cadherin code drives synthetic embryogenesis. The XEN cell cadherin code enables XEN cell sorting into a layer below ES cells, recapitulating the sorting of epiblast and primitive endoderm before implantation. The TS cell cadherin code enables TS cell sorting above ES cells, resembling extraembryonic ectoderm clustering above epiblast following implantation. Whereas differential cadherin expression drives initial cell sorting, cortical tension consolidates tissue organization. By optimizing cadherin code expression in different stem cell lines, we tripled the frequency of correctly formed synthetic embryos. Thus, by exploiting cadherin codes from different stages of development, lineage-specific stem cells bypass the preimplantation structure to directly assemble a postimplantation embryo.

Cadherins and protocadherins regulate cell adhesion forces in many different systems[1–4]. Cells expressing different types and levels of cadherins show differential cell–cell adhesion and sorting[1,5–8]. Moreover, synthetic genetic programs, in which distinct cell–cell contacts specify differential cadherin expression, can induce self-organization into multidomain structures and sequential assembly[9].

To determine the role of cadherins in the self-assembly of synthetic, so-called ETX embryos[10,11], we first re-analysed single-cell RNA sequencing (scRNA-seq) data that we published previously[10] to examine cadherin expression in the building blocks of ETX embryos: embryonic stem (ES), trophoblast stem (TS) and extraembryonic endoderm (XEN) cell lines (Fig. 1a). We found that E-cadherin (*Cdh1*) messenger RNA (mRNA) was equally abundant in ES and TS cells, whereas P-cadherin (*Cdh3*) was expressed only in TS cells, and K-cadherin (*Cdh6*) was expressed mainly in XEN cells (Fig. 1b,c). The differential expression of cadherins in ES, TS and XEN cells implies a potential role in driving the self-assembly of ETX embryos.

We then examined the expression of these three cadherins in cells dissociated from either ETX or natural embryos at successive stages (Fig. 1a). ES/epiblast, TS/trophectoderm and XEN/primitive endoderm lineages were defined by the expression of their respective markers and showed similar dynamics of cadherin expression in ETX and natural embryos (Fig. 1d and Extended Data Fig. 1a). In natural embryos, E-cadherin was expressed in all lineages from E4.5 to E6.5; P-cadherin expression was elevated only in trophectoderm after implantation (E5.5 and E6.5); and K-cadherin expression was elevated only in primitive endoderm before implantation (E4.5) when it sorts below the epiblast (Fig. 1d and Extended Data Fig. 1b). We verified that the corresponding proteins, similar to mRNA, were differentially expressed in ES or TS colonies (Extended Data Fig. 1c), day 4 ETX embryos and E5.5 natural embryos (Extended Data Fig. 1d). Therefore, XEN cells most resemble E4.5 primitive endoderm cells of the preimplantation embryo, whereas TS cells resemble extraembryonic ectoderm cells of the postimplantation embryo.

ES cells readily form chimeras with eight-cell-stage embryos and sort to the epiblast lineage. Given that E- and K-cadherin were differentially expressed in the epiblast and primitive endoderm of natural preimplantation embryos, we examined whether overexpression (OE) of these cadherins in ES cells would affect their subsequent sorting in the blastocysts of chimeras (Fig. 1e). We found that wild-type ES cells (*n* = 32 embryos) and ES cells overexpressing E-cadherin (*Cdh1* OE) contributed exclusively to the epiblast of the chimeras (*n* = 16 embryos) (Fig. 1f,g). In contrast, ES cells overexpressing K-cadherin (*Cdh6* OE) frequently contributed to the primitive endoderm (*n* = 16 embryos) (Fig. 1f,g and Extended Data Fig. 1e). These data are consistent with K-cadherin promoting primitive endoderm localization and E-cadherin promoting epiblast localization.

We noted that ES cells overexpressing P-cadherin (*Cdh3* OE) were excluded from the preimplantation embryo (*n* = 13 embryos) and sorted outside the trophectoderm (Fig. 1f,g and Extended Data Fig. 1f), consistent with low P-cadherin expression in all lineages of the blastocyst and elevated P-cadherin expression in the trophectoderm only in the postimplantation natural embryo.

To evaluate whether differential adhesion plays a role in ETX embryo self-assembly, we used atomic force microscopy (AFM) to determine the cell–cell adhesion of ES, TS and XEN cells in vitro (Fig. 2a,b). We found that the mean adhesion forces between ES–ES cell couples (1.94 ± 0.54 nN) and TS–TS cell couples (2.20 ± 0.85 nN) were significantly higher than those between

[1]Division of Biology and Biological Engineering, California Institute of Technology, Pasadena, CA, USA. [2]Mammalian Embryo and Stem Cell Group, Department of Physiology, Development and Neuroscience, University of Cambridge, Cambridge, UK. [3]The Francis Crick Institute, London, UK. [4]Department of Biochemistry, University of Cambridge, Cambridge, UK. ✉e-mail: magdaz@caltech.edu

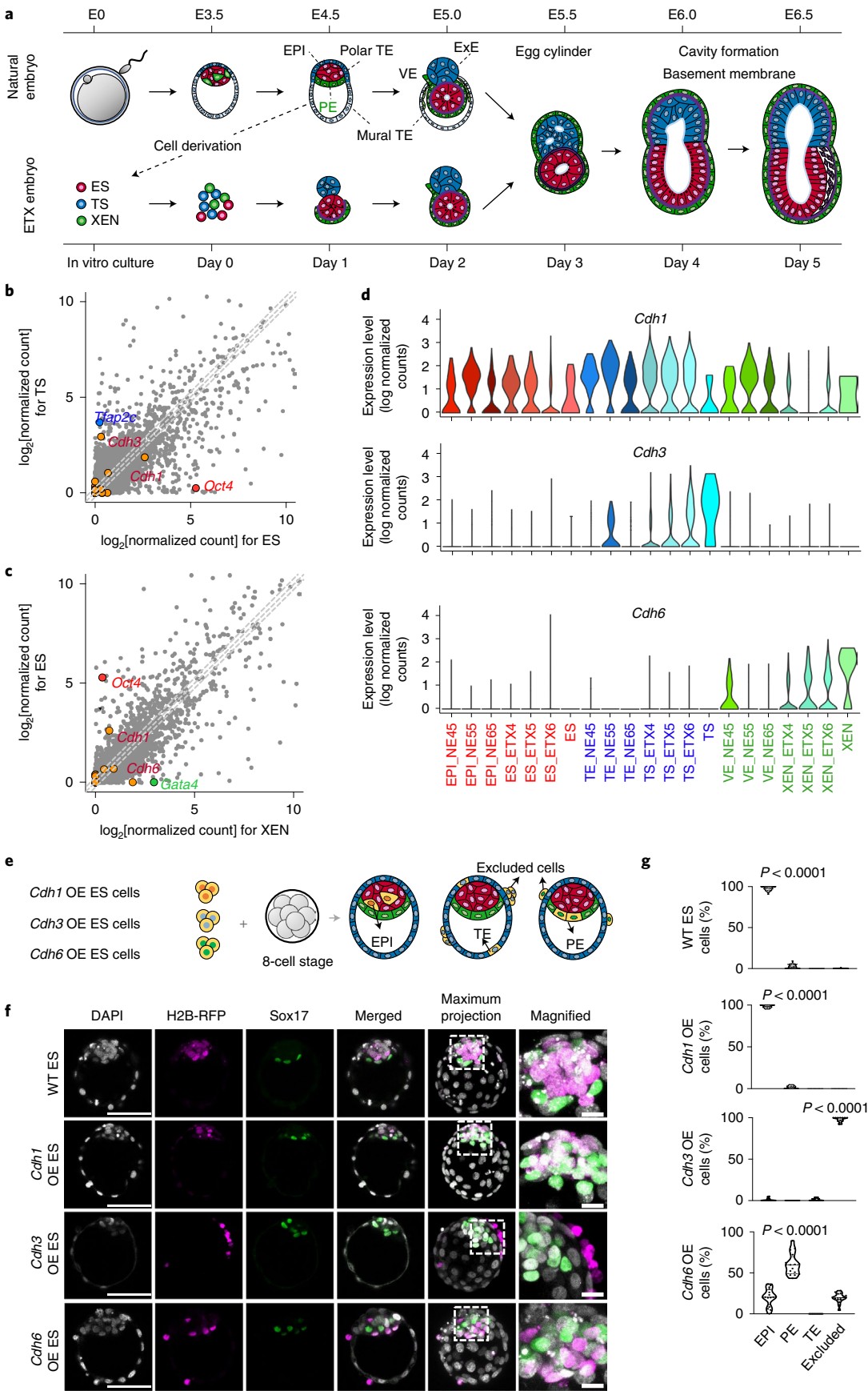

**Fig. 1 | Differential cadherin code in ETX and natural embryos. a**, Schematic showing self-organization and morphological transitions in natural and stem cell-derived (ETX) embryos. Red, epiblast (EPI) in the natural embryo and ES cells in the ETX embryo. Blue, trophectoderm (TE) in the natural embryo and TS cells in the ETX embryo. Green, primitive endoderm (PE) and visceral endoderm (VE) in the natural embryo and XEN cells in the ETX embryo. Purple, mesoderm. ExE, extra-embryonic ectoderm. **b**, Comparison of the average scRNA-seq read counts between ES and TS cells. Data points to the left (right) of the grey dashed lines represent transcripts enriched in TS (ES) cells by more than twofold. Points on the middle of the grey dashed line indicate equally expressed genes. **c**, Comparison of the average scRNA-seq read counts between XEN and ES cells. Data points to the left (right) of the grey dashed lines represent transcripts enriched in ES (XEN) cells by more than twofold. Points on the middle of the grey dashed line indicate equally expressed genes. In **b** and **c**, cadherin- and protocadherin-related transcripts are highlighted in orange. **d**, Violin plots showing *Cdh1* (top), *Cdh3* (middle) and *Cdh6* gene expression (bottom) from scRNA-seq in natural and ETX embryos at different stages. NE45, NE55 and NE65 represent natural embryos collected at day 4.5, 5.5 and 6.5. ETX4, ETX5 and ETX6 represent ETX embryos collected at day 4, 5 and 6. **e**, Schematic of chimera aggregation. Cadherin OE ES cells expressing H2B-RFP were aggregated with eight-cell-stage wild-type embryos. Their contribution to either EPI (red), PE (green), TE (blue) or excluded cells was assessed at E4.5. Orange, chimeric contribution. **f**, Chimeras stained for RFP (magenta), Sox17 (green) and DNA (DAPI; grey). Scale bars, 50 μm. The magnified images show the regions indicated by dashed boxes to the left (scale bars, 10 μm). The experiments were repeated three times. WT, wild type. **g**, Percentage of cells contributing to EPI, PE, TE or excluded cells in chimeras, as in **e**. The data are presented as violin plots. Each dot corresponds to an embryo. $n = 32$ embryos for wild-type ES chimeras (3365 cells in total), $n = 16$ embryos for *Cdh1* OE ES chimeras (1787 cells in total), $n = 13$ embryos for *Cdh3* OE ES chimeras (1574 cells in total) and $n = 16$ embryos for *Cdh6* OE ES chimeras (1894 cells in total). Statistical significance was determined by one-way ANOVA with a multiple comparison test. Numerical data are available as source data.

XEN–XEN cell couples ($0.55 \pm 0.11$ nN) or between ES–TS cell couples ($0.57 \pm 0.36$ nN), thus indicating a tendency for ES cells and TS cells to form homotypic associations. In addition, the adhesion forces between XEN–ES cell couples ($0.83 \pm 0.96$ nN) were greater than those between XEN–XEN cell couples ($0.55 \pm 0.11$ nN) or XEN–TS ($0.46 \pm 0.24$ nN) cell couples, suggesting that XEN cells have the highest affinity for ES cells (Fig. 2c). We also inferred adhesion forces from the contact angles between cells[12] (Fig. 2d). The contact angles at ES–ES, TS–TS and XEN–ES junctions were greater than the contact angles between ES–TS, XEN–XEN and TS–XEN cells (Fig. 2e and Extended Data Fig. 2a), in agreement with the AFM measurements.

To compare cell–cell contact angles in ETX and natural embryos, we employed the imaging surface analysis environment (ImSAnE) algorithm[13,14] to extract focal planes from three-dimensional (3D) stacks of E-cadherin-stained day 4 ETX and E5.5 natural embryos and unrolled these into 2D projections (Extended Data Fig. 2b). In agreement with cell contact angle measurements in stem cell doublets, we found that the homotypic contact angles were larger than the heterotypic contact angles in ETX and natural embryos (Extended Data Fig. 2c). We also noted that the contact angle between XEN–XEN and VE–VE cells at the surface of ETX embryos and natural embryos was close to 180° (Extended Data Fig. 2c,d), indicating a smooth boundary interface and reflecting the high relative tension along the interface after self-organization. Together, these measurements indicate differential cadherin expression and

differential adhesion between stem cells that build—and lineages that comprise—ETX embryos.

To examine the potential relationship between differential cadherin expression and differential adhesion, we measured adhesion forces between ES and TS cells and immobilized E-cadherin (*Cdh1*) or P-cadherin (*Cdh3*) substrate (Fig. 2f). ES cells exhibited higher adhesion with immobilized E-cadherin ($2.13 \pm 0.83$ nN) than with P-cadherin ($1.07 \pm 0.54$ nN), whereas TS cells displayed comparable adhesion forces with both E-cadherin ($2.02 \pm 0.89$ nN) and P-cadherin ($2.41 \pm 0.86$ nN). We then used RNA interference to knockdown (KD) cadherins in stem cells. E-cadherin KD reduced ES–ES adhesion fourfold. P- or E-cadherin KD similarly reduced TS–TS adhesion, suggesting that downregulation of one cadherin was sufficient to decrease the TS–TS adhesion force below a critical threshold (Fig. 2g and Extended Data Fig. 2e). Depletion of either E- or P-cadherin from XEN cells did not affect their homotypic adhesion (Fig. 2g). Thus, E-cadherin is required for the homotypic adhesion of ES cells, whereas both E- and P-cadherin are required for the homotypic adhesion of TS cells.

To assess whether the measured adhesion forces are sufficient to generate ETX embryos, we simulated assembly using the cellular Potts model (CPM)[15], re-sampling our AFM adhesion force measurements to provide parameters (Fig. 2h). This analysis showed that, among the many sorted configurations possible with three cell types, ETX-like structures were the most favoured (Fig. 2i) (see Supplementary Information).

**Fig. 2 | Differential adhesion force in ETX embryos. a**, Schematic showing cell–cell adhesion force measurement by AFM. **b**, The resulting force–distance curve, following the procedure depicted in **a**, enables quantification of the maximum adhesion force ($F_{max}$). **c**, $F_{max}$ for the indicated homotypic and heterotypic adhesions between three different cell types. The experiments were performed three times independently. Total measured cell pairs: $n = 60$ (ES–ES), $n = 177$ (TS–TS), $n = 101$ (XEN–XEN), $n = 124$ (ES–TS), $n = 148$ (XEN–TS) and $n = 134$ (XEN–ES). Statistical significance was determined by one-way ANOVA with a multiple comparison test. **d**, Schematics of weakly and strongly adherent cell pairs at force equilibrium. $\theta$ is the contact angle of the two adhering cells. **e**, Distribution of the measured contact angles at all cell–cell contacts. Total measured cell pairs: $n = 31$ (ES–ES), $n = 38$ (TS–TS), $n = 30$ (XEN–XEN), $n = 32$ (TS–ES), $n = 36$ (XEN–TS) and $n = 29$ (XEN–ES). $N = 3$ for all conditions. Statistical significance was determined by one-way ANOVA with a multiple comparison test. **f**, Adhesion forces between cells and different cadherins. Left, schematic showing cell–cadherin adhesion force measurement by AFM. Right, quantification of the results. $n = 42$ (ES–*E*-cadherin), $n = 35$ (ES–*P*-cadherin), $n = 41$ (TS–*E*-cadherin) and $n = 37$ (TS–*P*-cadherin). $N = 3$ for all of the conditions. Statistical significance was determined by unpaired two-tailed Student's *t*-test. **g**, $F_{max}$ for homotypic adhesion between the three different cell types after downregulation of *Cdh1* or *Cdh3*. $n = 60$ (WT ES–ES), $n = 18$ (*Cdh1* KD ES–ES), $n = 19$ (*Cdh3* KD ES–ES), $n = 177$ (wild-type TS–TS), $n = 20$ (*Cdh1* KD TS–TS), $n = 20$ (*Cdh3* KD TS–TS), $n = 101$ (wild-type XEN–XEN), $n = 19$ (*Cdh1* KD XEN–XEN) and $n = 19$ (*Cdh3* KD XEN–XEN). $N = 3$ for all conditions. Statistical significance was determined by one-way ANOVA with a multiple comparison test. **h**, Heatmap of the adhesion parameter matrix, generated by sampling measured AFM adhesion forces, which parameterizes the CPM. **i**, Bootstrapping procedure to infer the distributions of conformations under the CPM ($N = 498$). The schematic represents all of the possible sorted conformations, demonstrating that the ETX-like configuration is the most represented. Conformations observed at a frequency of <5% are grouped. MCS, Markov Chain Steps. In the box and whisker plots in **c** and **e**–**g**, the line inside the box indicates the median value and the error bars show the minimum and maximum values. Box edges indicate lower and upper quartile value. Numerical data are available as source data.

Next, we determined how the observed cadherin code affects the efficiency of ETX embryogenesis. Single-cell suspensions of ES, TS and XEN cells seeded into microwell plates assembled into multiple structures, of which 15.4% formed ETX structures recapitulating postimplantation embryo morphogenesis (Fig. 3a

and Supplementary Video 1). In contrast, 38.2% of structures had more than one ES compartment, 30.8% had more than one TS compartment and 12.8% had mislocalized XEN cells or lacked an outside XEN layer; we termed these missorted ETX structures (Fig. 3b–d). The proportion of correctly sorted ETX embryos

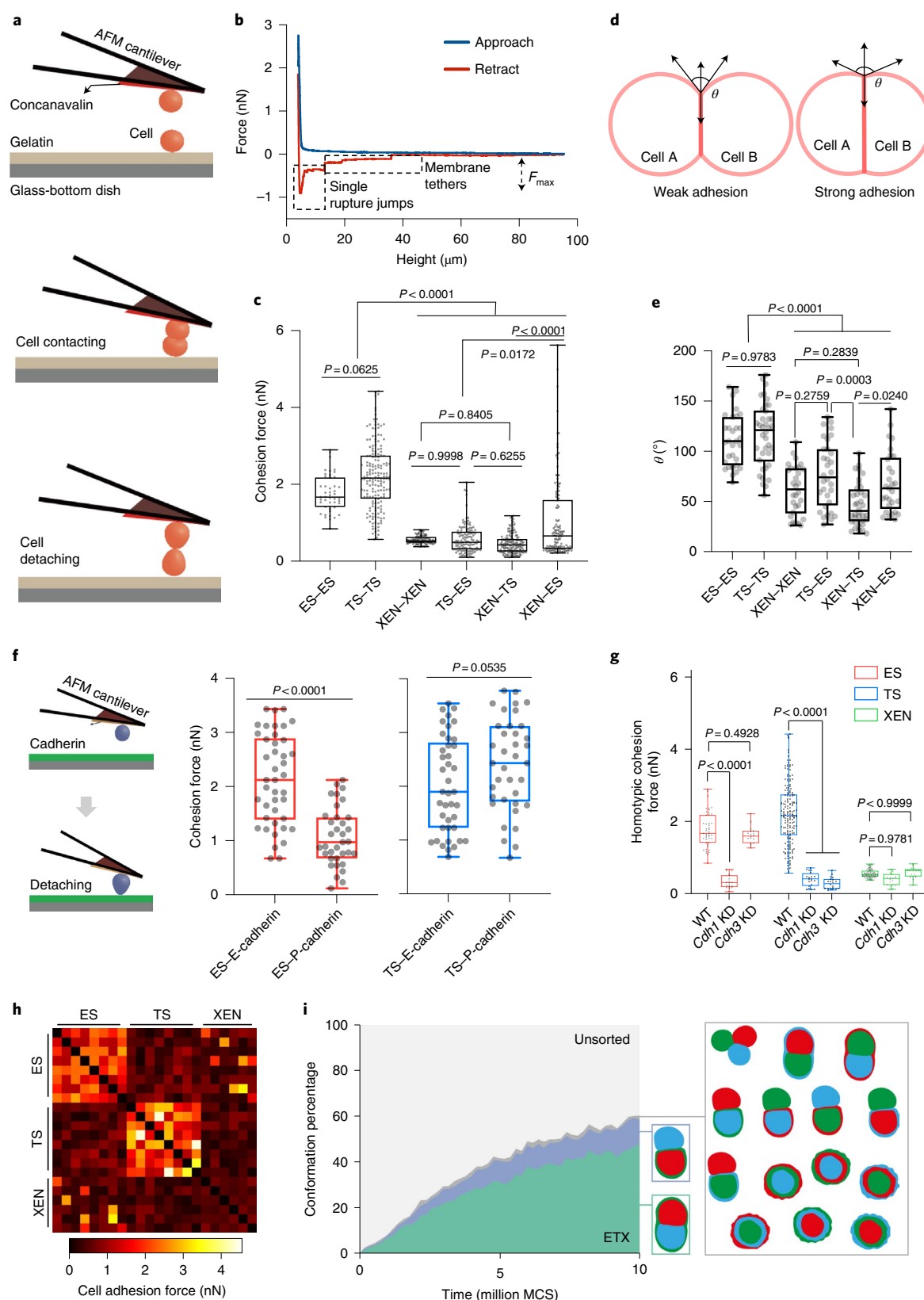

plateaued at 15% after the first day of culture (Extended Data Fig. 3a). Thus, the three cell types undergo a sorting phase within the first 24 h of seeding before becoming consolidated into compartments. We hypothesized that cells can no longer sort during the consolidation phase due to their low mobility. To test this, we filmed ETX embryo formation by time-lapse microscopy and tracked cell mobility (Extended Data Fig. 3b). This revealed all cell types to be mobile during the cell sorting stage, becoming relatively immobile during the tissue consolidation stage (Extended Data Fig. 3c,d).

XEN cell sorting fell into a particular two-phase pattern. XEN cells sorted efficiently; over 90% of XEN cells formed a monolayer, first enveloping ES cells and then spreading to cover TS cells. KD of E-cadherin (*Cdh1*) *or* K-cadherin (*Cdh6*), which are co-expressed in XEN cells, reduced the frequency of ETX embryos having a continuous XEN layer (Fig. 3e,f). However, XEN cells with OE of *Cdh1* or *Cdh6* often missorted within these compartments (Fig. 3e,f). Thus, an optimal balance of E-cadherin and K-cadherin contributes to the proper sorting of XEN cells in ETX embryos.

Differential adhesion cannot fully account for the ability of XEN cells to envelop the TS layer because the adhesion force between ES and XEN cells is larger than that between ES and TS cells, and XEN cells were found between the ES and TS compartments in approximately 10% of CPM simulations that sampled these data (Fig. 2i). The discrepancy between the predicted interfacial hierarchy for the sorted configuration in the ETX embryo and the measured differential adhesion force led us to hypothesize that the low number of XEN cells we used for making ETX embryos was insufficient to cover all ES cells during the sorting stage. To test this, we seeded ES and TS cells with between five and ten XEN cells and fixed the nascent structures at days 1 and 3. We found that low numbers of XEN cells first covered the ES cells only; subsequently, the TS cells enveloped the entire structure (Extended Data Fig. 3e,f). When we seeded approximately ten ES cells per structure, the XEN cells completely covered the ES cells, thereby excluding TS cells (Extended Data Fig. 3e,f), consistent with our measurements of differential adhesion.

Previous studies have reported a role for cortical stiffness in cell sorting, particularly in cell externalization[16–18], prompting us to consider whether cortical tension may influence the capacity of XEN cells to form their external monolayer. Indeed, our AFM[17] measurements indicated that cortical stiffness is lower in XEN cells than in either TS or ES cells (Extended Data Fig. 3g). To determine whether differences in cortical stiffness between the different stem cell types of ETX embryos were due to differential actomyosin activity, as in other systems[17,19,20], we measured cortical stiffness in the presence of blebbistatin (a myosin inhibitor[21]). Blebbistatin reduced the cortical stiffness of both ES and TS cells to the same level as in XEN

cells (Extended Data Fig. 3h). We also found that well-sorted ETX embryos treated with either blebbistatin or cytochalasin D (an actin depolymerizer[22]) at day 3 for 24 h—once the primary sorting phase was completed—failed to maintain efficient sorting compared with control ETX embryos (Extended Data Fig. 3h). Moreover, when we treated well-sorted ETX embryos with either blebbistatin or cytochalasin D for 24 h at day 3, once the primary cell sorting phase was complete, more than 80 and 85% of blebbistatin- and cytochalasin D-treated structures, respectively, failed to maintain sorting compared with 18% of control ETX embryos (Extended Data Fig. 3h).

To further test the role of cortical stiffness on XEN cell externalization, we used a CPM in which cortical stiffness can be tuned independently. We found that lower stiffness increased both the sorting efficiency and speed of XEN cell externalization (Extended Data Fig. 3i), suggesting that the softness of XEN cells is important for this event. Together, these data suggest that, in addition to the differential expression of distinct cadherins, cortical stiffness plays a role in the self-assembly of stem cells into ETX embryos.

Next, we examined the function of the cadherin code in ES and TS cells during ETX embryo assembly. KD of P-cadherin (*Cdh3*) in TS cells, but not in ES cells, resulted in TS mislocalization and disrupted ETX embryogenesis. Similarly, KD of E-cadherin (*Cdh1*) in ES cells, but not TS cells, disrupted ETX embryogenesis. ETX embryo formation still occurred following E-cadherin depletion from TS cells (Fig. 3g and Extended Data Fig. 4a), suggesting that differential expression of P-cadherin between ES cells and TS cells is sufficient to drive their sorting. We noticed that E-cadherin and P-cadherin showed different levels of expression in individual wild-type ES and TS cells, respectively (Extended Data Fig. 4b–d). We considered that subsets of wild-type stem cells with low cadherin expression compromise ETX embryo formation. Indeed, when we combined wild-type ES and XEN cells with either a P-cadherin OE subset or a P-cadherin KD subset of TS cells, P-cadherin KD TS cells mislocalized to the ES compartment. Similarly, when combining wild-type XEN and TS cells with either an E-cadherin OE subset or an E-cadherin KD subset of ES cells, we observed mislocalization of E-cadherin KD ES cells in the TS compartment (Extended Data Fig. 5a,b). Thus, populations of ES and TS cells with low E-cadherin and low P-cadherin expression, respectively, compromise sorting in ETX embryos. Strikingly, mixing E-cadherin OE ES cells and P-cadherin OE TS cells with wild-type XEN cells increased ETX embryogenesis efficiency by almost threefold from approximately 15% with wild-type stem cells to approximately 42% with the OE cells (Fig. 3h). The time course of the sorting of E-cadherin OE ES cells, P-cadherin OE TS cells and XEN cells revealed that around 30% of these structures were well-sorted 12 h after cell seeding compared with 6.8% of wild-type structures (Extended Data Fig. 5c,d).

**Fig. 3 | Differential cadherin code regulates self-organization in ETX embryos. a**, Representative images of the assembly of ETX embryos at different times. Scale bar, 50 μm. Blue, Tfap2c; green, Gata4; red, Oct4. **b**, Diversity of self-assembled structures collected at day 3. Scale bar, 100 μm. Staining as in **a**. **c**, Representative images of correctly sorted and missorted ETX structures after 3 d. The inset schematics show examples of the sorting outcomes. Scale bar, 100 μm. **d**, Pie chart showing the proportions of correctly sorted and missorted ETX structures at day 3. The 4000 structures analysed contained three different stem cell type. Four independent experiments were performed. **e**, Representative images of cell sorting resulting from combining *Cdh1* or *Cdh6* KD or OE XEN cells with wild-type ES and TS cells. Wild-type XEN cells provided the control. Scale bar, 100 μm. Staining as in **a**. **f**, Quantification of ETX structures with well-sorted or missorted XEN cells formed by XEN cells overexpressing (OE) *Cdh1* or *Cdh6* or KD for either *Cdh1* or *Cdh6*. Total numbers of structures: n = 470 (WT XEN), n = 282 (*Cdh1* KD XEN), n = 519 (*Cdh6* KD XEN), n = 326 (*Cdh1* OE XEN) and n = 281 (*Cdh6* OE XEN). N = 3. The data are presented as means ± s.d. Statistical significance was determined by one-way ANOVA with a multiple comparison test. **g**, Left, representative images of ETX structures of *Cdh1* and *Cdh3* KD ES and TS cells. Scale bar, 100 μm. Right, quantification showing well-sorted and missorted ETX embryos under the indicated conditions. Total numbers of structures: n = 4186 (control), n = 2940, (*Cdh1* KD ES), n = 2471 (*Cdh3* KD ES), n = 2407 (*Cdh1* KD TS) and n = 2151 (*Cdh3* KD TS). N = 3. The data are presented as means ± s.d. Statistical significance was determined by one-way ANOVA with a multiple comparison test. **h**, Left, representative images of the ETX structures formed by combining *Cdh1* OE ES cells (red) with *Cdh3* OE TS cells (blue) and wild-type XEN cells (green). Middle, magnified images indicating enlarged well-sorted ETX structures, as indicated by the white arrows to the left. Scale bars, 100 μm. Right, quantification of the well-sorted ETX structures, n = 3451 (control) and n = 2348 (*Cdh1* and *Cdh3* OE) structures were selected from five independent experiments. The data are presented as means ± s.d. Statistical significance was determined by unpaired two-tailed Student's *t*-test. The experiments were repeated four times in **a**–**c** and three times in **e**. Numerical data are available as source data.

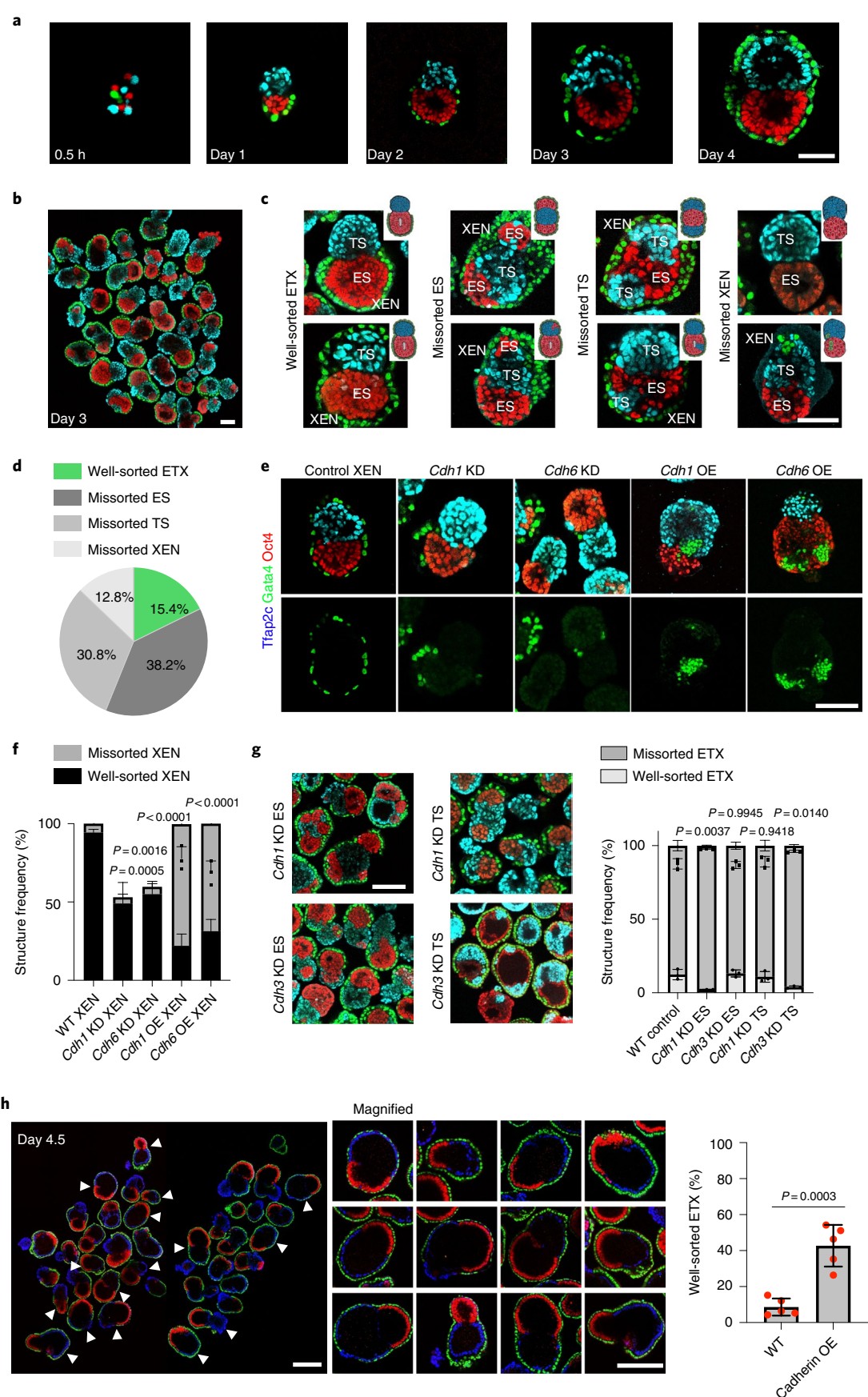

Thus, the sorting rate is increased following cadherin OE, as suggested in simulations[23]. Together, these results indicate that variable E-cadherin expression in ES cells and P-cadherin expression in TS cells limits the efficiency of ETX embryo formation.

As implantation-stage embryo morphogenesis requires both lumenogenesis and basement membrane formation, we wanted to determine whether cadherin-enhanced ETX embryo self-organization also improved these events. ETX embryos generated from wild-type stem cells formed a central lumen—corresponding to the lumen of the epiblast rosette at implantation[24,25]—within the ES compartment by day 2. By day 3, multiple lumens developed in the TS compartment, corresponding to the multiple lumens of the E5.5 extraembryonic ectoderm, and these unified into a single cavity between days 4 and 5, as in natural development by E6.0 (Fig. 4a). Such a single unified cavity formed in over 90% of properly sorted ETX embryos but in fewer than 5% of ETX structures with missorted ES and TS cells (Extended Data Fig. 6a). Moreover, ETX structures with missorted XEN cells lacked cavities entirely (Extended Data Fig. 6b). Importantly, proper sorting and amniotic cavity-like formation were observed in only 9% of structures built from wild-type cells but in 40% of structures built from E-cadherin OE ES cells, P-cadherin OE TS cells and wild-type XEN cells (Fig. 4b,c). Moreover, the structures formed from cadherin OE cells, and the cavities within them, were longer than in ETX embryos built from wild-type cells (Extended Data Fig. 6c,d) after 3 d in culture. Together, this indicates that E- and P-cadherin OE in ES and TS cells, respectively, promotes cavity formation in ETX embryos.

Lumenogenesis requires signalling from the basement membrane, produced by the visceral endoderm[24,26]. Accordingly, we found that ETX structures with missorted XEN cells, which lacked a cavity, also failed to establish a basement membrane (Extended Data Fig. 6e). A continuous laminin-containing basement membrane was detected in 78% of structures built from E-cadherin OE ES cells, P-cadherin OE TS cells and XEN cells (Fig. 4d) but in only 45% of structures made from wild-type ES, TS and XEN cells (Fig. 4e). Thus, elevated expression of E- and P-cadherin in ES and TS cells, respectively, increases the successful formation of basement membrane, lumen and correctly sorted ETX embryos (Fig. 4f).

Our findings shed light on the remarkable self-assembly of stem cells into synthetic embryos[10,11,27–32]. We show that this requires a cadherin code that, through strong homotypic interactions, sorts ES and TS cells into distinct compartments. In contrast, heterotypic interactions enable XEN cells to first surround ES and then TS cells. Although XEN cells have a cadherin code resembling preimplantation primitive endoderm, they nevertheless attain the ability to support synthetic postimplantation morphogenesis (Extended Data Fig. 6f). These differences between natural and ETX embryos highlight the distinct use of common rules between biological development and bioengineering. Synthetic embryo assembly utilizes these codes in a distinct way: XEN cells use the preimplantation code of primitive endoderm to sort in a layer below ES cells, whereas TS cells use the postimplantation code of extraembryonic ectoderm to sort as a cluster above ES cells.

The outcome of cell sorting has been modelled previously by considering cell-specific differences in interfacial energies maximizing the most energetically favourable cell interfaces[15,33–36]. Disparity in interfacial energy was considered to reflect adhesion differences, with cadherins being the best-characterized effectors[1,37], as espoused in the differential adhesion hypothesis (DAH)[33,38]. In accord, we now show that cell sorting is driven in ETX embryos by increased cadherin-mediated homotypic interactions in relation to heterotypic interactions. The later development of the differential interfacial tension hypothesis (DITH)[16,17,39–41] invoking the role of differential cortical tension in sorting resonates with our findings on XEN cell externalization in self-assembly. Together, our observations support the balance between adhesion and tension (DAH versus DITH) as in biophysical models of cell sorting. However, incomplete ES–TS sorting still results in local order, emphasizing a need for global-scale sorting to fully recapitulate natural morphogenesis. DAH and DITH only account for local sorting to form homotypic clusters of ES and TS cells, as seen even in missorted structures. For complete sorting, ETX embryos must escape from locally correct neighbourhoods within globally incorrect patterns to explore alternative conformations. If cells remain in local minima before cell sorting is complete, structures will remain missorted.

The importance of the cell type-specific cadherin code is illustrated by our finding that established wild-type stem cell lines show heterogeneous cadherin expression, with some subsets below the threshold required to support proper sorting. Elevating E-cadherin in ES cells and P-cadherin in TS cells substantially improves ETX embryogenesis efficiency (Fig. 4f). This identifies a broader challenge in synthetic biology; namely, characterizing and ameliorating the impacts of heterogeneity in stem cell lines—factors that remain for the most part undefined but reported[35,42–44]. Such heterogeneity might affect the distribution of cell–cell cohesive properties within the same cell population[45], confounding the hierarchy of interactions necessary to drive self-organization of other organoid structures[46,47]. Thus, the principles of self-organization that we now describe can provide a means for increasing the efficiency of formation of different types of organoids by modulating interactions and the physical properties of cells.

## Online content

Any methods, additional references, Nature Research reporting summaries, source data, extended data, supplementary information, acknowledgements, peer review information; details of

---

**Fig. 4 | Correct self-organization is necessary for proper morphogenesis. a**, Time course of the assembly of ETX embryos stained to reveal E-cadherin (monochrome), Oct4 (red) and Gata4 (green). The bottom row of images are magnifications of the images above and show E-cadherin staining around a nascent cavity, as indicated by the dashed yellow lines. The dashed green line indicates the boundary between the ES and XEN compartment. Scale bar, 5 μm. **b**, Representative images showing Oct4 (red), Gata4 (green), E-cadherin (monochrome) and DAPI (grey) staining in day 4 cadherin OE ETX structures formed by combining E-cadherin OE ES cells with P-cadherin OE TS cells and wild-type XEN cells. ETX structures formed by combining wild-type cells were used as a control. Scale bars, 100 μm. **c**, Comparison and quantification of joined cavity formation in cadherin OE and control ETX structures. n = 361 (control group) and n = 253 (cadherin OE group). N = 5 for each condition. The data are presented as means ± s.d. Statistical significance was determined by unpaired two-tailed Student's t-test. **d**, Representative image showing Oct4 (red), Gata4 (green), laminin (monochrome) and DAPI (blue) staining in day 4 cadherin OE ETX structures formed by combining E-cadherin OE ES cells with P-cadherin OE TS cells and wild-type XEN cells. ETX structures formed by combining wild-type cells were used as a control. Scale bars, 100 μm. **e**, Quantification of the structures that contained continuous or discontinuous laminin. n = 40 ETX structures per condition. N = 3. The data are presented as means ± s.d. Statistical significance was determined by unpaired two-tailed Student's t-test. **f**, Self-organization principles in stem cell-derived ETX embryos. Differential expression of E-, K- and P-cadherins enables the sorting of ES (epiblast-like), XEN (VE-like) and TS (TE-like) stem cells. Wild-type ES cells with low E-cadherin expression and wild-type TS cells with low P-cadherin expression exhibited detrimental global sorting efficiency. This could be overcome by overexpressing E-cadherin in ES cells and P-cadherin in TS cells to increase the efficiency of ETX embryo formation. Proper morphogenesis, including cavity formation, basement membrane formation (purple) and symmetry breaking can only be observed in well-sorted structures. Numerical data are available as source data.

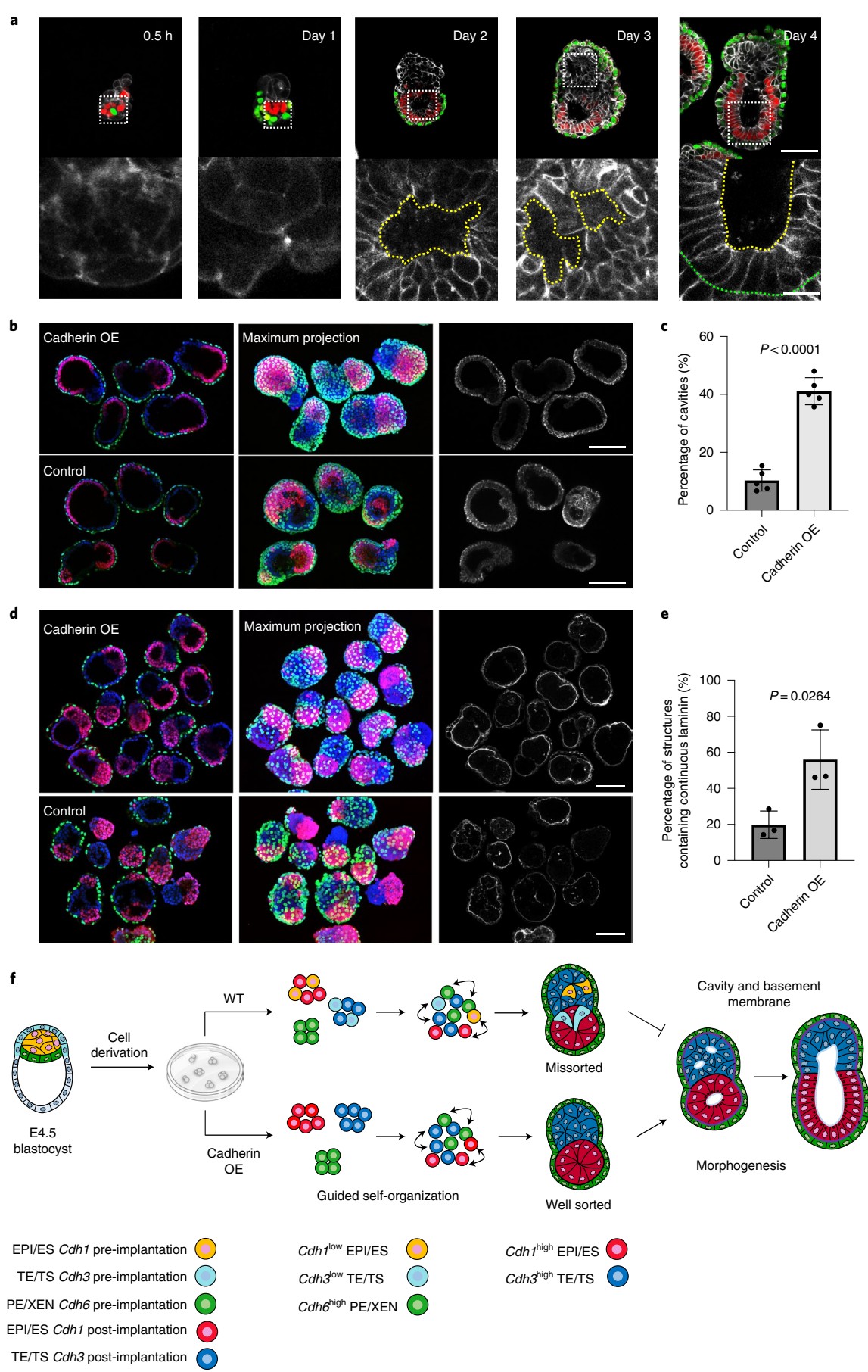

author contributions and competing interests; and statements of data and code availability are available at https://doi.org/10.1038/s41556-022-00984-y.

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

## Methods

**Cell culture.** All cells were cultured at 37 °C in 20% $O_2$ and 5% $CO_2$ and passaged once they had reached 70% confluency. Cells were tested weekly for *Mycoplasma* contamination by PCR.

ES cells were cultured on a 0.1% gelatin-coated plate in N2B27 medium with 1 μM MEK inhibitor PD0325901, 3 μM GSK3 inhibitor CHIR99021 and 10 ng ml⁻¹ leukaemia inhibitory factor. The N2B27 medium comprised a 1:1 mix of Dulbecco's modified Eagle medium (DMEM)/F12 (21331-020; Thermo Fisher Scientific) and Neurobasal-A (10888-022; Thermo Fisher Scientific) media supplemented with 0.5% vol/vol N2 (17502048; Thermo Fisher Scientific), 1% vol/vol B27 (10889-038; Thermo Fisher Scientific), 100 μM β-mercaptoethanol (31350-010; Thermo Fisher Scientific), 1% vol/vol penicillin–streptomycin mix (15140122; Thermo Fisher Scientific) and 1% vol/vol GlutaMAX (35050-061; Thermo Fisher Scientific).

TS cells (wild type) were cultured on mitomycin C (M4287; Sigma–Aldrich)-treated CF1 mouse embryonic fibroblasts (MEFs) in TSF4H medium with RPMI 1640 (M3817; Sigma–Aldrich) containing 20% foetal bovine serum (FBS; 35-010-CV; Thermo Fisher Scientific), 2 mM L-glutamine, 0.1 mM β-mercaptoethanol, 1 mM sodium pyruvate, 1% penicillin–streptomycin (M7167; Sigma–Aldrich), 25 ng ml⁻¹ FGF4 (5846-F4; R&D Systems) and 1 μg ml⁻¹ heparin (H3149; Sigma–Aldrich).

TS cells (*Cdh3* OE) were cultured on 10 μg ml⁻¹ laminin-coated plates in TX medium, with 50 ng ml⁻¹ IL11 (50117-MNCE; Sino Biological), 50 ng ml⁻¹ activin (Qk001-ActA-100; Qkine), 25 ng ml⁻¹ Bmp7 (PeproTech; 120-03P-10μg), 5 nM lysophosphatidic acid (Santa Cruz Biotechnology; sc-201053) and 200 nM 8Br-cAMP (B 007-500; BIOLOG Life Science Institute). TX medium was made from a 1:1 mix of DMEM and F12 media (21331-020; Thermo Fisher Scientific) with 19.4 μg ml⁻¹ insulin (342106; Sigma–Aldrich), 64 μg ml⁻¹ L-ascorbic-acid (A4403; Sigma–Aldrich), 14 ng ml⁻¹ sodium selenite (S5261; Sigma–Aldrich), 543 μg ml⁻¹ sodium bicarbonate (S5761; Sigma–Aldrich), 10.7 μg ml⁻¹ holo-transferrin (T4132; Sigma–Aldrich), 1% penicillin–streptomycin (M7167; Sigma–Aldrich), 25 ng ml⁻¹ FGF4 (5846-F4; R&D Systems), 2 ng ml⁻¹ TGF-β1 (100-21 C; PeproTech) and 1 μg ml⁻¹ heparin (H3149; Sigma–Aldrich). After selection, *Cdh3* OE TS cells were cultured on MEF in TSF4H medium.

XEN cells were cultured on gelatin-coated plates in 70% MEF-conditioned IDG medium (c-IDG). c-IDG medium comprised DMEM (21969; Gibco) containing 12.5% FBS (35-010-CV; Thermo Fisher Scientific), 2 mM GlutaMax (35050-038; Gibco), 0.1 mM 2-mercaptoethanol (31350-010; Gibco), 0.1 mM nonessential amino acids (11140-035; Gibco), 1 mM sodium pyruvate (11360-039; Gibco), 0.02 M HEPES (15630080; Gibco) and 1% penicillin–streptomycin (15140122; Gibco)

MEF cells were cultured on gelatin-coated plates in DMEM medium (41966; Thermo Fisher Scientific) supplemented with 15% FBS (35-010-CV; Thermo Fisher Scientific), penicillin–streptomycin (15140122; Thermo Fisher Scientific), GlutaMAX (35050061; Thermo Fisher Scientific), MEM nonessential amino acids (11140035; Thermo Fisher Scientific), sodium pyruvate (11360070; Thermo Fisher Scientific) and 100 μM β-mercaptoethanol (31350-010; Thermo Fisher Scientific).

**Mouse embryos.** Mice were maintained according to national and international guidelines. All experiments were regulated by the Animals (Scientific Procedures) Act 1986 Amendment Regulations 2012 following ethical review by the University of Cambridge Animal Welfare and Ethical Review Body. Experiments were approved by the Home Office. Animals were inspected daily and those that showed health concerns were culled by cervical dislocation. Six-week-old female CD-1 mice were used in all of the animal experiments. All experimental mice were free of pathogens and were on a 12 h light/12 h dark cycle, with unlimited access to water and food. The temperature in the facility was controlled and maintained at 21 °C.

**Generation of cell lines.** The experiments were performed using mouse E14 wild-type ES cells[48] (derived in the laboratory of M.Z.-G.). The H2B-CFP ES cells were a gift from M. Elowitz, the wild-type TS cells were a gift from J. Nichols, the EGFP TS cells were a gift from J. Rossant, the wild-type XEN cells were a gift from E. Na and the H2B-RFP XEN cells were derived from wild-type XEN cells. *Cdh1* and *Cdh6* OE ES cells were generated from E14 wild-type ES cells, the *Cdh3* OE TS cells were generated from wild-type TS cells and the *Cdh1* and *Cdh6* OE XEN cells were generated from wild-type XEN cells (see below).

To generate cadherin OE ES, TS or XEN cell lines, 0.5 μg of a super piggyBac transposase expression vector (PB210PA-1; System Biosciences) and 2 μg *Cdh1*-pHygro, *Cdh3*-pHygro or *Cdh6*-pHygro plasmid were co-transfected into cells using Lipofectamine. Cells were passaged for 24 h after transfection and subjected to selection in medium containing 50 μg ml⁻¹ hygromycin (10687010; Thermo Fisher Scientific) for 1 week. A similar approach was used to generate the stable nuclear reporter H2B-RFP XEN cell line.

**Cloning.** Cloning procedures were performed using Gateway technology (Thermo Fisher Scientific). The fragment of interest (*Cdh1*, *Cdh3* or *Cdh6*) was amplified by PCR to introduce attB sites. These fragments were cloned into the pDONR221 vector (a gift from J. Silva) using BP clonase II (11789020; Thermo Fisher Scientific). The fragment of interest (*Cdh1*, *Cdh3* or *Cdh6*) was subcloned into a

pHygro vector containing a hygromycin resistance cassette for expression in stem cells. The recombination reaction was carried out using LR Clonase II (11791100; Thermo Fisher Scientific).

PiggyBac-based expression plasmids for *Cdh1*, *Cdh3* or *Cdh6* were generated by PCR amplification of the respective genes in pENTR-*Cdh1* (49776; Addgene), *Cdh3* (Myc-DDK tagged) (MR227345; OriGene Technologies) or *Cdh6* (mouse-tagged ORF clone) (MG222740; OriGene Technologies) with the oligos listed in Supplementary Table 1.

**Small interfering RNA.** Cells were transfected with 25 nM small interfering RNA (siRNA) directed against *Cdh1* (1027418-SI00946631), *Cdh3* (1027418-SI02666440) or *Cdh6* (1027418-SI00946967) (Qiagen) siRNA or with control scrambled siRNA (Qiagen) using Lipofectamine 3000 transfection reagent according to the manufacturer's instructions. Cells were harvested at 72 h post-transfection and assayed by quantitative PCR (qPCR).

**Flow cytometry analysis.** Single ES and TS cell suspensions were collected, fixed in 4% paraformaldehyde and permeabilized for 30 min at room temperature using 0.3% Triton X-100 and 0.1% glycine. Cells were then incubated with anti-E-cadherin (1:200; 13-1900; Thermo Fisher Scientific) or P-cadherin antibody (1:100; sc-1501; Santa Cruz Biotechnology) for overnight incubation at 4 °C in blocking buffer (phosphate-buffered saline (PBS) solution plus Tween 20) (PBST) containing 10% FBS). Cells were washed twice in PBST and then incubated with secondary antibody (1:500 dilution) in blocking buffer at room temperature for 1–2 h. Cells were then analysed by quantitative flow cytometry (BD Biosciences) and the intensity profiles of E- and P-cadherin were plotted using FlowJo software (version 10.7.1) (https://www.flowjo.com).

**Cell doublets experiment.** To measure cell–cell contact angles, 1200 dissociated ES, TS or XEN cells were mixed in pairs and seeded onto AggreWell plates (34411; STEMCELL Technologies) pretreated with rinsing solution (07010; STEMCELL Technologies). Cells were centrifuged at 100g for 3 min. After 1 h incubation at 37 °C, cells were collected and fixed for immunostaining.

**Cadherin-coated surface preparation.** To measure the adhesion forces between cells and cadherin-coated surfaces, we followed a previous study[49]. Briefly, gold-coated glass cover slips (AU.0100; Platypus) were cleaned in argon plasma for 30 s and subsequently functionalized by immersion in thiol solution (P50757; Sigma–Aldrich) for 16 h, then rinsed with EDTA-buffer (15575-020; Thermo Fisher Scientific) to remove excess thiol. The cover slips were subsequently incubated with 10 μg ml⁻¹ recombinant E-cadherin (8875-EC-050; R&D Systems) or P-cadherin (761-MP-050; R&D Systems) for 12 h at 4 °C. Before making force measurements, the cadherin-coated surfaces were washed with HEPES (15630106; Thermo Fisher Scientific) and activated by incubation in the same buffer for 30 min.

**Adhesion force measurement.** Cell–cell or cell–cadherin adhesion forces were measured using an atomic force microscope (Bruker NanoScope) coupled to a confocal microscope (TCS SP5II; Leica). Tipless silicon nitride cantilevers were V shaped, with nominal spring constants (60 pN nm⁻¹; NP-0; Veeco Instruments). The atomic force microscope cantilevers were plasma cleaned before functionalization with concanavalin A, as described previously[17,49]. The system was calibrated in cell-free medium at 37 °C before each experiment by measuring the deflection sensitivity on a glass surface, allowing the cantilever spring constant to be determined in situ. Before loading the sample, the sample stage movement was calibrated using NanoScope software (version 6.13). Before measurements, cells were dissociated with TrypLE (12604013; Thermo Fisher Scientific) and resuspended in HEPES-buffered cell culture medium (15630056; Thermo Fisher Scientific). Cell suspensions were loaded into the atomic force microscope sample chamber and a single cell was captured by pressing the cantilever onto the cell with a contact force of 500 pN for 1 min. The cell was lifted from the surface and allowed to establish firm adhesion on the cantilever for 5 min. To measure the cell–cell adhesion force, the captured cell was lowered to contact with another single cell cultured on a gelatin-coated glass-bottom Petri dish (FD35; WPI). To measure the cell–cadherin adhesion force, the captured cell was lowered to contact cadherin-coated cover slips. The approach and retraction speeds were kept constant at 10 μm s⁻¹ with a contact force of 2 nN. Three force curves were acquired for each cell. The captured cell was left to recover for 3 min between different adhesion force measurement cycles before it was adhered to the surface in a different position. Before and after every single measurement, we checked that our probing cell remained on the cantilever by direct observation. Maximal cell adhesion forces as well as the single rupture force step height were extracted from retrace curves using JPK IP software.

**Cell cortical stiffness measurements.** The stiffness of cells was measured using an atomic force microscope (Bruker NanoScope) coupled to a confocal microscope (TCS SP5II; Leica), as described previously[50,51]. The point-and-shoot procedure (NanoScope software; Bruker) was used to measure cell stiffness. All cells were kept in $CO_2$-independent cell culture medium during the measurement.

A fluorescent 10 µm polystyrene bead (Invitrogen) was glued to silicon nitride cantilevers with nominal spring constants of $0.06 \, N \, m^{-1}$ (NP-S type D; Bruker). Indentations were performed using the single force option with a total indentation depth of 50–100 nm. To obtain cell stiffness values from force curves, PUNIAS software was used as described previously[50,51]. Multiple force displacement curves (at five different locations) were fitted to the Hertz model to calculate cell cortical stiffness (Young's modulus).

**Stem cell-derived ETX embryo generation.** ETX embryos were generated as described previously[10]. Approximately 6000–7000 ES cells, 15,000–19,000 TS cells and 5000–6000 XEN cells were added dropwise into AggreWell plates having 1200 microwells in one well (34411; STEMCELL Technologies). The microwells were treated with rinsing solution (07010; STEMCELL Technologies). Cells were centrifuged at 100$g$ for 3 min. 1.5 ml c-IDG medium containing 7.5 nM ROCK inhibitor (72304; STEMCELL Technologies) was added dropwise to each well. On the following day (day 1), 1 ml medium was removed gently from each well and replaced with 1 ml fresh c-IDG medium without ROCK inhibitor. This step was repeated once to fully remove the ROCK inhibitor. On day 2, 1 ml c-IDG medium was replaced with 1 ml fresh medium. On day 3, the media was replaced with IVC1 medium[10,24,52]. IVC1 medium comprises advanced DMEM/F12 (21331-020; Gibco) supplemented with 20% (vol/vol) FBS, 2 mM GlutaMax, 1% vol/vol penicillin–streptomycin, 1× ITS-X (51500-056; Thermo Fisher Scientific), 8 nM β-estradiol, 200 ng ml$^{-1}$ progesterone and 25 mM $N$-acetyl-L-cysteine.

**Immunofluorescence.** Natural embryos, stem cell-derived structures or stem cells were fixed in 4% paraformaldehyde (15710; Electron Microscopy Sciences) for 20–30 min at room temperature, washed twice in PBST (containing 0.05% Tween 20) and permeabilized for 30 min at room temperature in 0.3% Triton X-100 and 0.1% glycine. Primary antibody incubation was performed overnight at 4 °C in blocking buffer (PBST containing 10% FBS). The following day, samples were washed twice in PBST and then incubated with secondary antibody (1:500) in blocking buffer at room temperature for 1–2 h. Embryos were transferred to PBST drops in oil-filled optical plates before confocal imaging.

The following primary antibodies were used: Tfap2c (1:200; AF5059; R&D Systems), Brachyury (1:200; AF2085; R&D Systems), Gata4 (1:500; 36966; Cell Signalling Technology), Laminin (1:500; L9393; Sigma–Aldrich), Oct4 (1:500; sc-5279; Santa Cruz Biotechnology), E-cadherin (1:200; 13-1900; Thermo Fisher Scientific) and P-cadherin (1:100; sc-1501 (Santa Cruz Biotechnology) or MS-1741 (Fisher Scientific)). The following secondary antibodies from Thermo Fisher Scientific were used: Alexa Fluor 488 Donkey anti-Mouse (1:500; A-21202), Alexa Fluor 488 Donkey anti-Goat (1:500; A-11055), Alexa Fluor 488 Donkey anti-Rat (1:500; A-21208), Alexa Fluor 568 Donkey anti-Rabbit (1:500; A-10042), Alexa Fluor 568 Donkey anti-Mouse (1:500; A-10037), Alexa Fluor 647 Donkey anti-Goat (1:500; A-21447) and Phalloidin (1:200; A30104). Detailed information of the used antibodies is provided in Supplementary Table 1.

**RNA extraction and real-time qPCR.** Total RNA was extracted from cells using TRIzol Reagent (15596-026; Invitrogen). Real-time qPCR was performed with SYBR Green PCR Master Mix (4368708; Applied Biosystems) and StepOnePlus Real-Time PCR System (Applied Biosystems). The fold change in mRNA expression was determined using the ΔΔCt method with Gapdh as an endogenous control. For the qPCR primers used, see Supplementary Table 1.

**scRNA-seq sample preparation and dissociation.** Natural and ETX embryos were transferred to Falcon tubes, washed with PBS and incubated in TrypLE Express (12604013; Gibco) for 15 min at 37 °C to dissociate them into single cells. If clumps remained, the incubation was extended for an additional 5 min at 37 °C and the sample pipetted further. Samples were filtered to remove large clumps, centrifuged at 200$g$ for 5 min and resuspended in PBST (containing 0.02% Tween 20) and then processed for encapsulation, as previously reported[29,53]. For E5.5 embryos, one litter of 12 embryos was dissociated together. A total of 15 ETX embryos were dissociated for sequencing. Cells in culture were dissociated into a single-cell suspension using TrypLE Express (12604013; Gibco) and multiplexed using MULTI-seq lipid-modified oligos before running on two 10X Genomics lanes using single-cell 3′ version 3 reagents as reported[54].

**scRNA-seq analysis.** A previously submitted and filtered scRNA-seq dataset comprising the ETX and natural embryos was downloaded from the Gene Expression Omnibus repository (GSE161947)[29]. The count matrix was loaded into Seurat version 3 (ref. [55]), the fraction of counts mapping to mitochondrial genes was computed and the object was then log-normalized to a scale factor of 10,000. The 2,000 most variable genes were computed, the object was scaled and the percentage of mitochondrial counts was regressed out. Dimensional reduction was performed with principal component analysis and the data were projected on a uniform manifold approximation and projection low-dimensional space using 20 principal components. The embryonic, endoderm and trophectoderm lineage identity was pooled from previous annotations[29], corresponding to clusters with high *Dnmt3b*, *Gata4* and *Lamb1* and *Cdx2* and *Gata2* expression levels, respectively. The average expression was computed using the average

expression function and the latter was log$_2$ normalized. The expression levels of cadherins and protocadherins were subsequently plotted on a heatmap. The uniform manifold approximation and projection plots were directly plotted using the methodology described recently[29] after keeping the ETX and natural embryo samples only and re-computing the neighbourhood graph (five neighbours and 30 principal components; code at https://github.com/fhlab/scRNAseq_inducedETX).

**Time-lapse imaging.** To perform time-lapse imaging, cells were seeded on Gri3D PEG-hydrogel dishes with glass bottoms (SUN Bioscience) and imaged under a spinning-disc microscope (3i) with a Zeiss EC Plan-NEOFLUAR 20×/0.5 objective in a humidified chamber at 37 °C with 5% $CO_2$. The structures were imaged every 5–10 min by collecting image stacks of 10 µm z-planes. Images were processed using SlideBook 5.0 (3i). Raw data were processed using the open-source image analysis software Fiji. For single-cell tracking, Imaris image analysis software (Bitplane) was used.

**Quantification and statistical analysis.** *Criteria for selecting ETX embryos.* Egg cylinder structures with one TS-derived compartment and one ES-derived compartment, covered by an outside XEN-derived visceral endoderm-like monolayer were considered to be well-sorted ETX embryos for analysis. Structures that did not fulfil these criteria were considered to be missorted ETX structures. Structures containing all three types of cells were collected and counted for quantification.

*Image data acquisition, processing and quantification.* Fluorescence images were acquired using an inverted Leica SP8 confocal microscope (LEICA software LAS X; Leica Microsystems) with a Leica FLUOTAR VISIR 25× or 40× objective. Images were acquired with 0.5–3.0 µm z-separation. To screen entire structures, the tile-scan imaging mode with automatic image stitching of the SP8 confocal microscope was used. All images were analysed and processed using Fiji software (http://fiji.sc). For digital quantifications and immunofluorescence signal intensity graphs, laser power and detector gain were maintained constant to permit quantitative comparisons of different experimental conditions within a single experiment.

To evaluate cell–cell contact angles in 3D ETX and natural embryos, ImSAnE[13,14] was employed to extract planes of the embryos from 3D stacks of E-cadherin and unroll them into a two-dimensional projection. Different lineages were indicated by different nuclear markers during analysis. Geometric observables as well as general distortions in projections can be correctly quantified using built-in correction methods in MATLAB[14].

**Numerical simulations.** *CPM.* A CPM[15] was used to infer the predicted distributions of conformations given measurements of cell adhesion from AFM, as well as to determine the roles of cortical stiffness on the self-organization of ETX embryos. We parameterized adhesion strengths using cohesion forces between pairs of cell types, which were directly measured by AFM (Supplementary Table 2). For each simulation, we sampled this distribution to build the adhesion ($J$) matrix. Specifically, for a given element in this matrix, we sampled (with replacement) the set of AFM cohesion forces measured between pairs of cell types (for example, ES–ES, ES–TS and so on), performed around 500 times to establish an ensemble of $J$ matrix samples. Each $J$ matrix sample was used to perform a CPM simulation, generating an ensemble distribution of conformations over time. Simulations evolve via a stochastic minimization of an energy function (see equation (1) in the Supplementary Information) that accounts for both differential affinity and other physical properties of cells. Simulations were scored at each time point for being one of the 16 possible sorted configurations (Fig. 2i) by determining whether each cell type was enveloping and/or contiguous (see Supplementary Information for details). To test for the importance of softness in XEN cell externalization, we varied the cortical stiffness parameter for XEN cells ($\lambda_P^{XEN}$) and repeated the above simulation procedure.

**Statistics and reproducibility.** Statistical tests were performed using GraphPad Prism (versions 8.0 and 7.0a) software (with the exception of the analysis of sequencing data). Data with a Gaussian distribution were analysed using a two-tailed unpaired Student's $t$-test (two groups) or one-way analysis of variance (ANOVA) (multiple groups) with Tukey's multiple comparison test. Significant differences in the variance were taken into account using Welch's correction. Data that did not have a Gaussian distribution were analysed using a Mann–Whitney $U$-test (two groups) or Kruskal–Wallis test (multiple groups) with Dunn's multiple comparison test. For all quantifications, a minimum of three independent experiments were performed. The in vitro cell experiments were not randomized as it was not necessary. For experiments with chemical inhibitors, samples were randomly allocated to control and experimental groups. Embryos were randomly allocated to control and experimental groups for the in vivo experiments. Data collection and analysis were not performed blind to the conditions of the experiments. No statistical method was used to predetermine sample sizes. Sample sizes were determined based on previous experimental experience. The sample sizes used to derive statistics are provided in each figure caption. No data

were excluded from the analyses. Sequencing data were analysed using standard programs and packages. Significance levels are shown in each graph.

**Reporting summary.** Further information on research design is available in the Nature Research Reporting Summary linked to this article.

## Data availability

Previously published scRNA-seq data that were re-analysed here are available under accession code GSE161947. All other data supporting the findings of this study are available from the corresponding author upon reasonable request. Source data are provided with this paper.

## Code availability

The source code used for the numerical simulations is available on GitHub at https://github.com/jakesorel/CPM_ETX_2022.

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

## Acknowledgements

This work was supported by the Wellcome Trust (207415/Z/17/Z), an European Research Council advanced grant (669198), a National Institutes of Health R01 (HD100456-01A1) grant, the National Institutes of Health Pioneer Award (DP1 HD104575-01), the Tianqiao and Chrissy Chen Institute for Neuroscience and Shurl and Kay Curci Foundation grants to M.Z.-G. E.S.-V. is supported by a Pew Latin America fellowship. M.B. is supported by a Caltech Postdoctoral Fellowship. We thank the Life Science Foundation, members of the M.Z.-G. laboratory and A. Winkel for invaluable comments and suggestions.

## Author contributions

M.B. and M.Z.-G. conceived of the idea. M.B. developed the methodology and performed the investigation. J.D.J., F.H. and M.B. performed the bioinformatics analyses. J.C.-S., S.S. and M.T. performed the modelling. E.S.-V. and M.B. performed the chimera experiments. A.L.C. contributed to experimental design, pilot experiments and schematics. D.-Y.C. and J.C.-S. provided experimental assistance. M.Z.-G. supervised the study. D.M.G. co-supervised the study. M.B., J.C.-S., D.M.G. and M.Z.-G. wrote the paper.

## Competing interests

We would like to disclose that we have filed a patent for this study. The applicants and inventors for this Patent are Min Bao and Magdalena Zernicka-Goetz. The patent was filed on September 2, 2022 by Caltech. This patent pertains to and covers the "Differential adhesion and tension guided formation of stem cell derived embryos". The Patent was filed under the following number: 63/403685. The remaining authors declare no competing interests.

## Additional information

**Extended data** is available for this paper at https://doi.org/10.1038/s41556-022-00984-y.

**Correspondence and requests for materials** should be addressed to Magdalena Zernicka-Goetz.

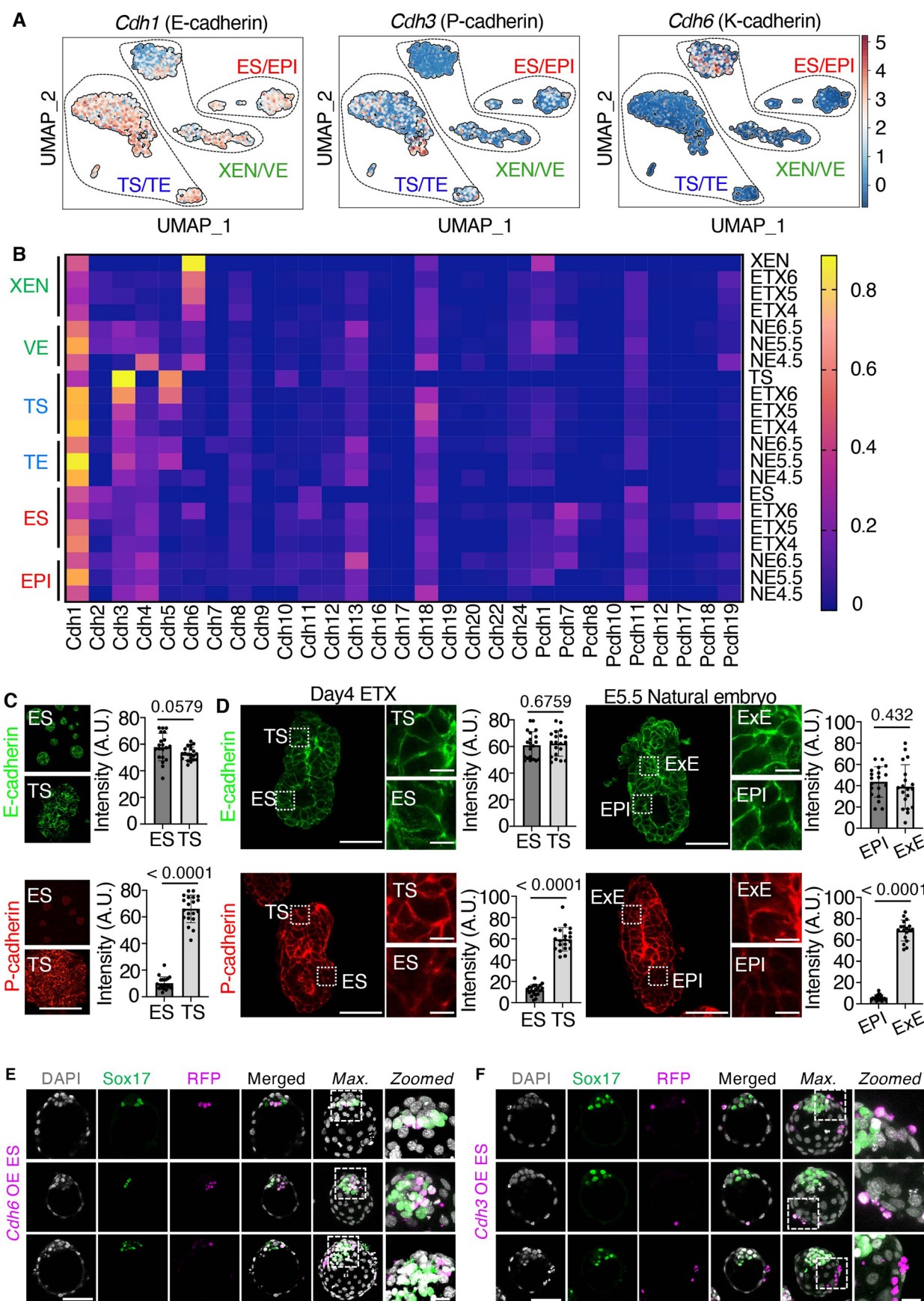

**Extended Data Fig. 1 | See next page for caption.**

**Extended Data Fig. 1 | Differential cadherin code in ETX-embryo and natural embryo. (a**) UMAP dimensional reduction shows *Cdh1*, *Cdh3* and *Cdh6* expression profile in different clusters as indicated by dashed lines. Each dot represents a single cell that is color-coded by sample type. (**b**) Heatmap showing average expression of cadherin and protocadherin related genes revealed by scRNA-seq in natural embryos (NE, n=50) collected at 4.5, 5.5 and 6.5 days after fertilization and well-sorted ETX embryos (n=50) at 4, 5, 6 days of culture. (**c**) Colonies of cultured ES and TS cells stained to reveal E-cadherin (green) and P-cadherin (red). Quantifications showing the mean intensity (A.U.) of E-cadherin or P-cadherin at cell-cell junctions. 20 colonies from 3 different experiments were selected for quantification. Scale bars represent 100 μm. Data are presented as mean ± SD. Statistics calculated by unpaired two-tailed Student's *t* test. (**d**) Natural embryos (E5.5) and ETX embryos (Day 4) stained to reveal E-cadherin (green) and P-cadherin (red). Magnified insets show E- or P-cadherin staining in ExE and EPI compartments in natural embryos, TS and ES compartments in ETX embryos. Quantifications showing the mean intensity (A.U.) of E-cadherin or P-cadherin at cell-cell junctions. n=20 ETX embryos and n=19 natural embryos were used for quantification. Data are presented as mean ± SD. Statistics calculated by unpaired two-tailed Student's *t* test. Scale bars represent 100 μm (main Figure) and 20μm (inset). (**e**) Representative images of E4.5 chimeras (8-cell stage embryos aggregated with *Cdh6* or (**f**) *Cdh3* OE ES stained for RFP (magenta), Sox17 (green), and DAPI (grey). Experiments were repeated 3 times. Scale bars represent 50 μm. Zoomed images are of regions indicated by dashed lines (scale bars represent 10 μm). Source numerical data are available in source data.

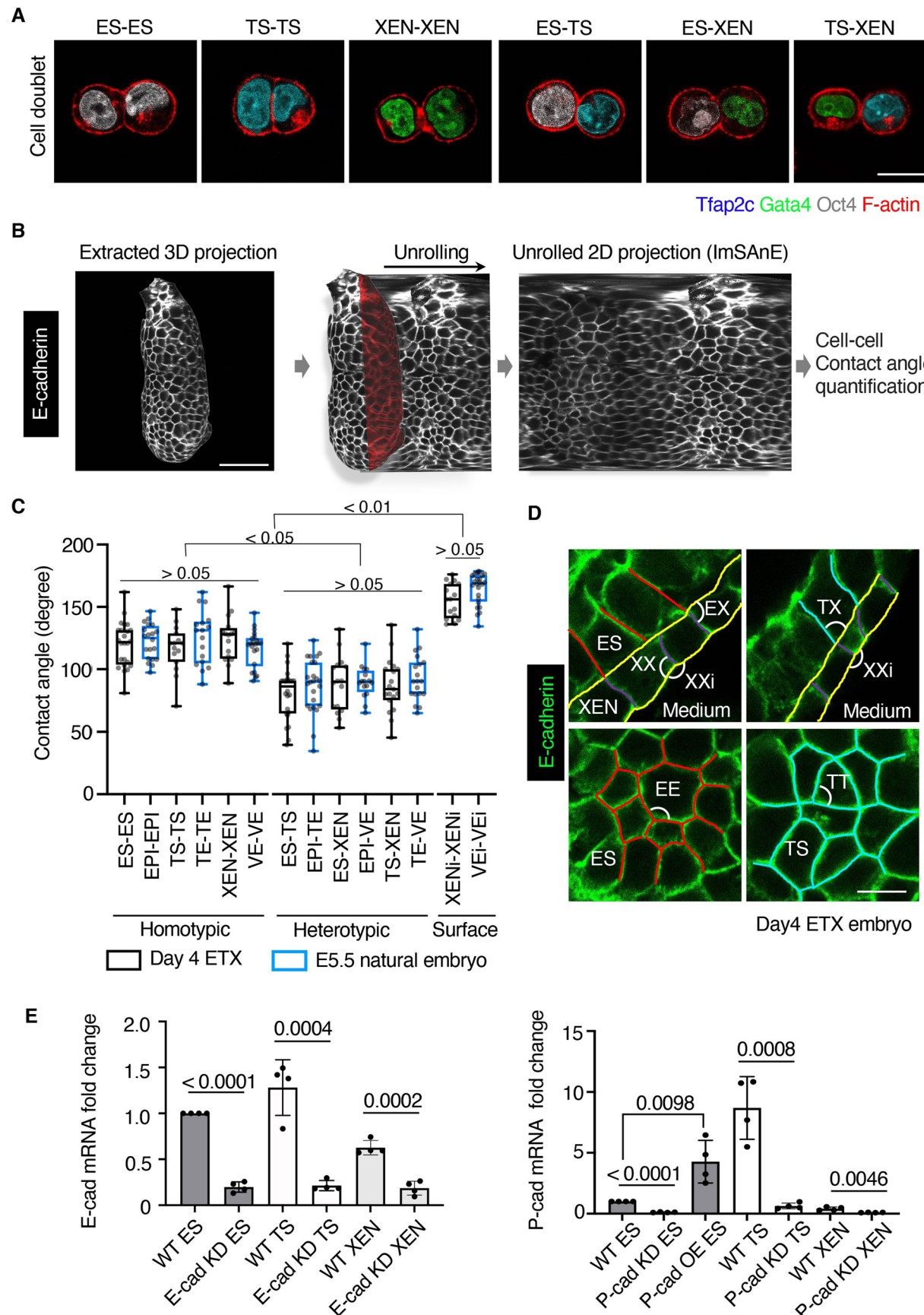

**A**

ES-ES   TS-TS   XEN-XEN   ES-TS   ES-XEN   TS-XEN

Cell doublet

Tfap2c Gata4 Oct4 F-actin

**B**

E-cadherin

Extracted 3D projection  →Unrolling→  Unrolled 2D projection (ImSAnE)  → Cell-cell Contact angle quantification

**C**

Contact angle (degree)

< 0.01
< 0.05
> 0.05    > 0.05    > 0.05

ES-ES EPI-EPI TS-TS TE-TE XEN-XEN VE-VE | ES-TS EPI-TE ES-XEN EPI-VE TS-XEN TE-VE | XENi-XENi VEi-VEi

Homotypic | Heterotypic | Surface

☐ Day 4 ETX   ☐ E5.5 natural embryo

**D**

E-cadherin

EX ES XX XXi XEN Medium | TX XXi Medium

EE ES | TT TS

Day4 ETX embryo

**E**

E-cad mRNA fold change

< 0.0001      0.0004      0.0002

WT ES  E-cad KD ES  WT TS  E-cad KD TS  WT XEN  E-cad KD XEN

P-cad mRNA fold change

0.0098      0.0008
< 0.0001              0.0046

WT ES  P-cad KD ES  P-cad OE ES  WT TS  P-cad KD TS  WT XEN  P-cad KD XEN

**Extended Data Fig. 2 | See next page for caption.**

**Extended Data Fig. 2 | Differential adhesion force in ETX embryos.** (**a**) Representative images of cell doublets for homotypic and heterotypic cell pairs (green, Gata4; cyan, Tfap2c; gray, Oct4; red, F-actin). Experiments were repeated 3 times. Scale bar is 10 µm. (**b**) Use of ImSAnE 'Unrolling' algorithm to project 3D E-cadherin staining stacks onto a 2D plane. Cell contact angles were quantified using built-in correction methods. Geometric observables as well as generally distortions in projections can be correctly quantified. Scale bar is 100 µm. (**c**) Cell-cell contact angle measurements based on ImSAnE method in day 4 ETX and E5.5 natural embryos. Total measured cell pairs in ETX embryos: ES-ES: n = 24; TS-TS: n = 15; XEN-XEN: n = 16; ES-TS: n = 24; XEN-TS: n = 19; XEN-ES: n = 16; XENi-XENi: n = 16. Total measured cell pairs in natural embryos: EPI-EPI: n = 20; TE-TE: n = 17; VE-VE: n = 22; EPI-TE: n = 24; EPI-VE: n = 16; TE-VE: n = 19; VEi-VEi: n = 24. Data are presented as box-whisker plots, black line inside the box indicates the median value and the error bar shows min to max value. Statistics calculated by one-way ANOVA with a multiple comparison test. (**d**) Enlargement of the boundary area in a day4 ETX embryo stained with E-cadherin, with homotypic contacts highlighted in blue (TS-TS), red (ES-ES) and purple (XEN-XEN), and the heterophilic boundary interface in yellow. Angles formed at tricellular junctions between different types are indicated: EX, TX and ET, angles between heterotypic contacts (ES-XEN, TS-XEN and ES-TS); EE, TT and XX, angles between homotypic contacts (ES-ES, TS-TS and XEN-XEN). XXi indicates contact angles of XEN cells at cell-medium interface. Experiments were repeated 6 times. Scale bar represents 20 µm. (**e**) E-cadherin and P-cadherin mRNA expression in cells after downregulation of E- or P-cadherin by RNAi, scrambled siRNA was used as a control. P-cadherin mRNA expression in ES cells after overexpression of P-cadherin. N = 4 for all conditions. Data are presented as mean ± SD. Statistics calculated by unpaired two-tailed Student's *t* test. Source numerical data are available in source data.

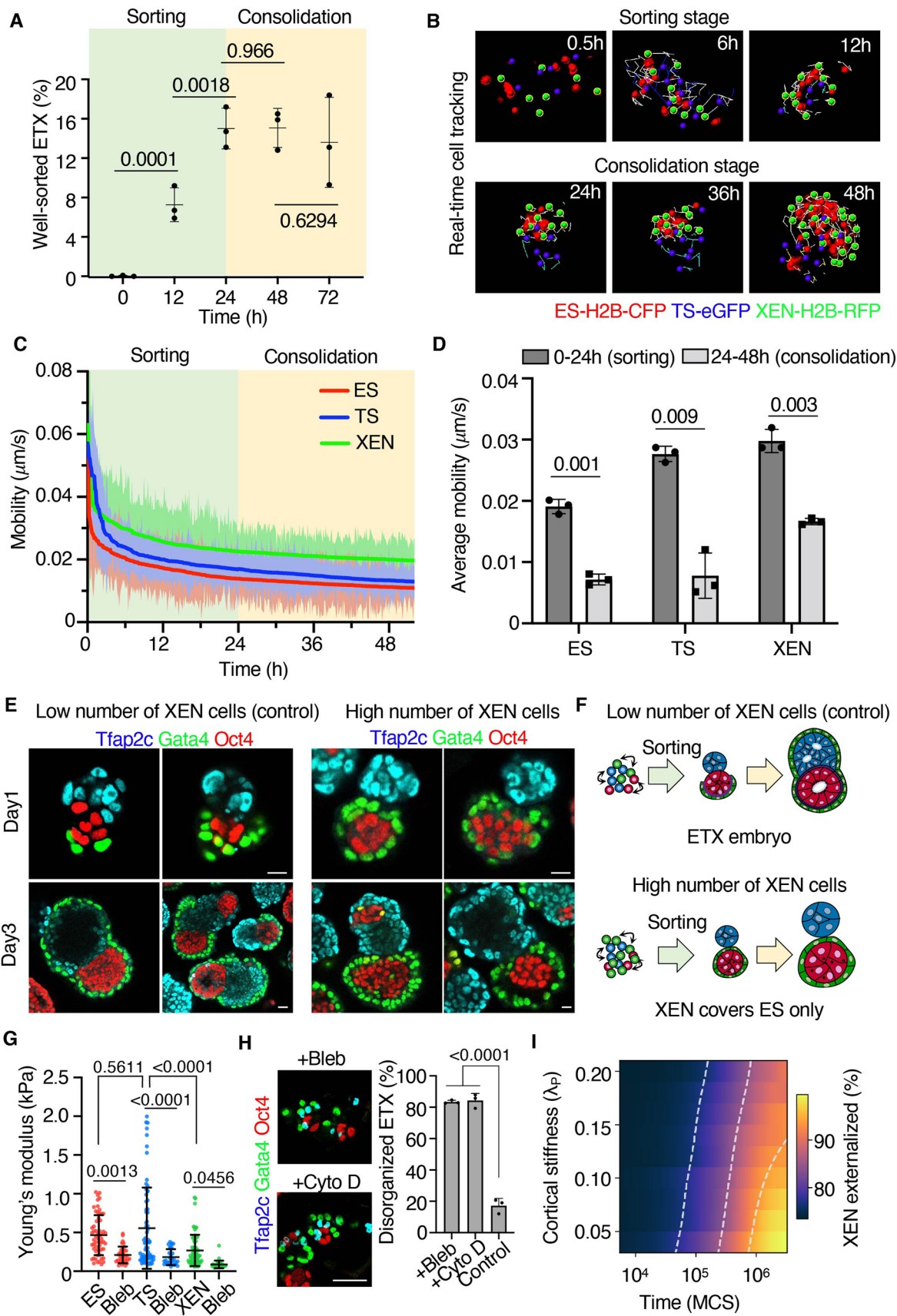

**Extended Data Fig. 3 | See next page for caption.**

**Extended Data Fig. 3 | Differential cadherin code and cortical tension regulate self-organization in ETX embryos.** (**a**) Time course of formation of correctly-sorted ETX embryos following seeding. 0.5-h: 0/515 structures; 12-h: 79/1292 structures; 24-h: 160/1074 structures; 48-h: 134/888; 72-h: 93/702 structures. N = 3 for each condition. Data are presented as Mean ± SD. Statistics calculated by unpaired two-tailed Student's *t* test. (**b**) Live cell imaging and tracking. H2B-RFP-XEN (green), H2B-CFP-ES (red) and EGFP-TS (blue) were overlaid with Imaris cell-tracking spheres. (**c**) Quantification of mobility for different types of cells during self-organization. Data are presented as Mean ± SD at different time points. (**d**) The bar graph shows the average mobility for different cell types during self-organization at different time ranges after cell seeding. Data are presented as Mean ± SEM. 12 structures from 3 independent experiments were imaged for quantification. Statistics calculated by unpaired two-tailed Student's *t* test. (**e**) Examples of structures made from low (control) and high number of XEN cells, stained at day-1 and day-3. Experiments were repeated 3 times. Scale bar, 10 μm. (**f**) Schematic of morphological transitions when using low and high number of XEN cells. (**g**) Cortical stiffness measurements for indicated cell types before and after treatment with Blebbistatin (Bleb). Total measured cell numbers for each condition: ES: n = 58; ES + Bleb: n = 34; TS: n = 68; TS + Bleb: n = 31; XEN: n = 68; XEN + Bleb: n = 35. Data are presented as Mean ± SD. Statistics calculated by ANOVA with a multiple comparison test. (**h**) Day 3 well-sorted ETX embryos were cultured with either blebbistatin, cytochalasin D or DMSO (control) during consolidation stage for 24 hrs and immuno-stained to reveal the indicated markers. Quantification shows the percentage of disorganized ETX structures. n = 84 (Bleb treated), n = 83 (Cyto D treated) and n = 75 (control), N = 3 for each condition. Data are presented as Mean ± SD. Statistics calculated by unpaired two-tailed Student's *t* test. Scale bar, 100 μm. (**i**) CPM modelling shows the effect of XEN cell stiffness ($\lambda_p$) on externalization efficiency (N = 474). The sorting efficiency calculated for each time-point is plotted as a heat-map, overlayed with contours (white dotted lines). Source numerical data are available in source data.

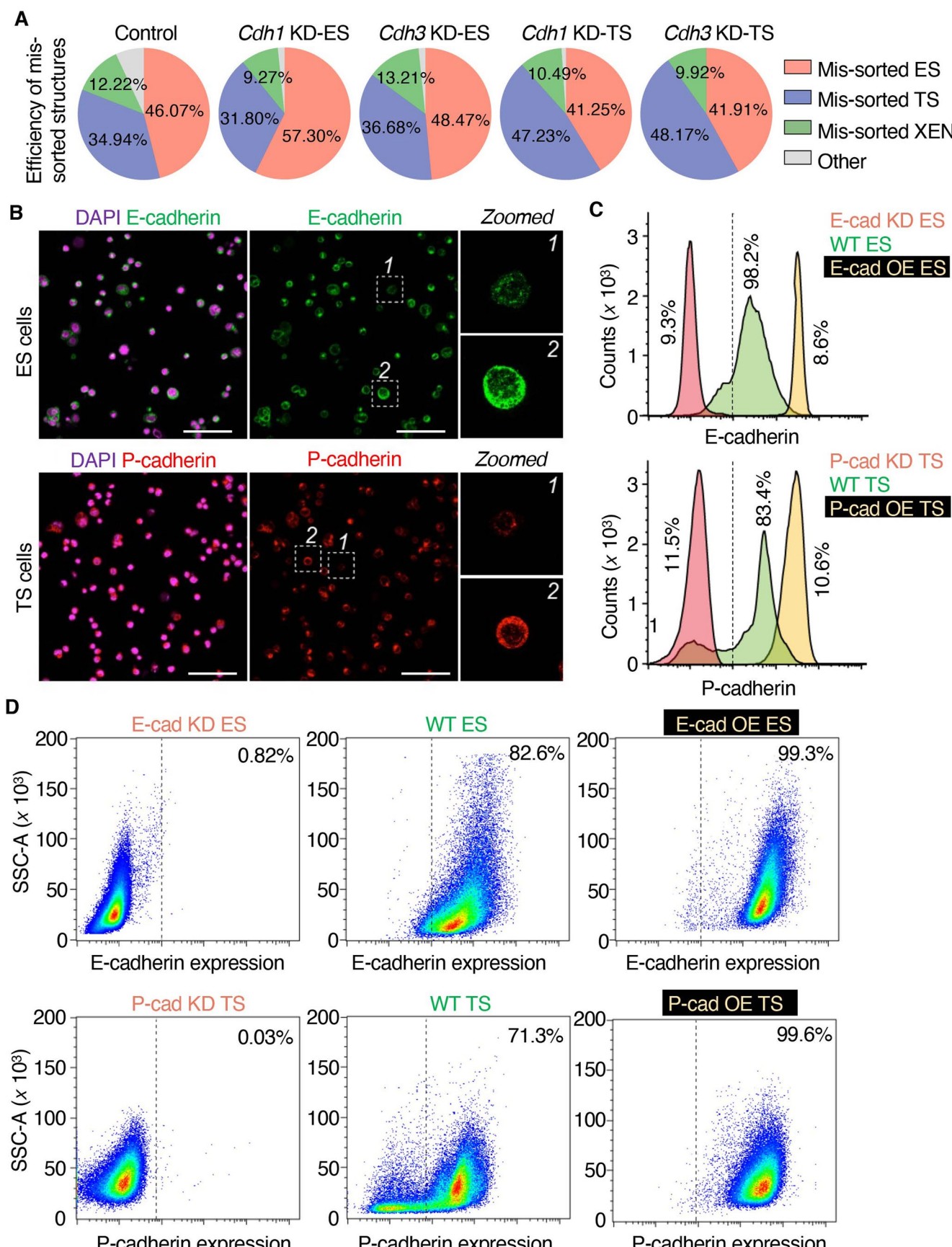

**Extended Data Fig. 4 | See next page for caption.**

**Extended Data Fig. 4 | Cadherin heterogeneity within the same cell population in ETX embryos.** (**a**) Pie charts show different mis-sorted ETX embryos under the indicated conditions. n = 4186 (control), n = 2940 (*Cdh1*-KD ES), n = 2471 (*Cdh3*-KD ES), n = 2407 (*Cdh1*-KD TS), n = 2151 (*Cdh3*-KD TS) structures were collected from 3 independent experiments for quantification. (**b**) Immuno-staining of ES (upper) and TS (lower) cells to reveal E-cadherin (green) and P-cadherin (red), respectively. Nuclei (purple) are stained by DAPI. Scale bars represent 100 μm. Zoomed images are of regions indicated by dashed lines. Experiments were repeated 5 times. (**c**) Flow cytometric analysis of E-cadherin in wild-type ES cells, E-cadherin knockdown ES cells and ES cells over-expressing E-cadherin (upper). Flow-cytometric analysis of P-cadherin in wild-type TS cells, P-cadherin knockdown TS cells and TS cells over-expressing P-cadherin (lower). CV (coefficient of variation) values shown against peak values in plots. (**d**) Top: FACS profiles for E-cadherin in E-cadherin KD, WT and E-cadherin OE ES cells, Bottom: FACS profiles for P-cadherin in P-cadherin KD, WT and P-cadherin OE TS cells.

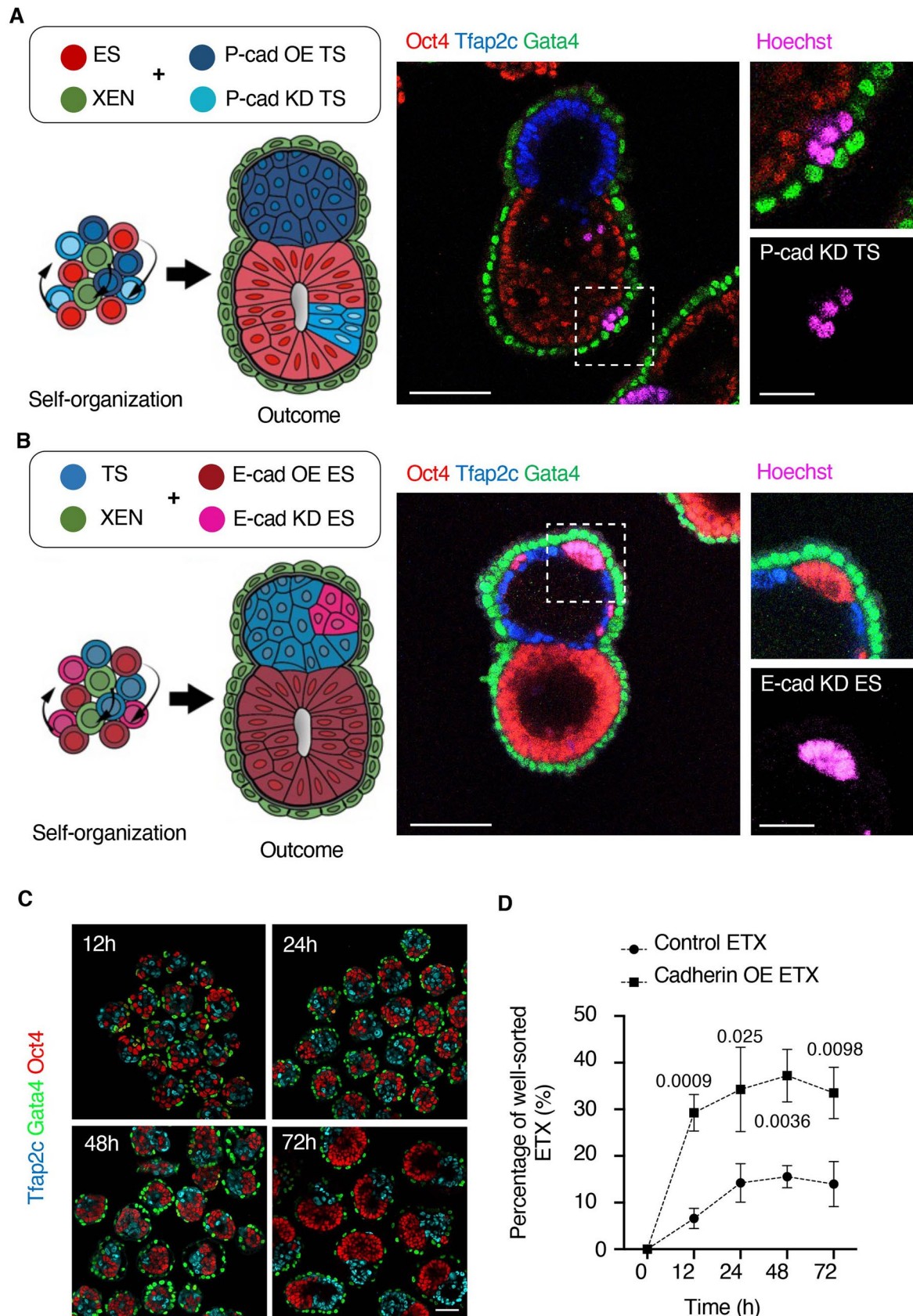

**Extended Data Fig. 5 | See next page for caption.**

**Extended Data Fig. 5 | Cadherin heterogeneity affects cell positioning in ETX embryos.** (**a**) Schematics and representative images for assembled day 4 ETX-embryos from TS cells (upper) or (**b**) ES cells (lower) overexpressing (OE) or knocked-down (KD) for the indicated cadherins. Experiments were repeated 6 times. Scale bar represents 100 μm. P-cadherin-overexpressing TS cells and E-cadherin-overexpressing ES cells were pre-stained with Hoechst to distinguish them from cadherin knockdown cells. Scale bar represents 40 μm in zoomed panels. (**c**) Examples of structures made from E-cadherin-OE-ES, P-cadherin-OE-TS and XEN cells, stained at different time points to reveal ES cells (Oct4, red), TS cells (Tfap2c, blue) and XEN cells (Gata4, green). Scale bar represents 100 μm. (**d**) Quantification shows time course of formation of correctly-sorted ETX embryos following seeding. Control: 12 h: 24/332 structures; 24 h: 83/531 structures; 48 h: 71/448 structures; 72 h: 51/378 structures. Cadherin OE: 12 h: 80/276 structures; 24 h: 139/385 structures; 48 h: 136/374 structures; 72-h: 151/455 structures. N = 3 for all conditions. Data are presented as Mean ± SD. Statistics calculated by unpaired two-tailed Student's *t* test. P values indicate significance between control and Cadherin OE ETX at the same time point. Source numerical data are available in source data.

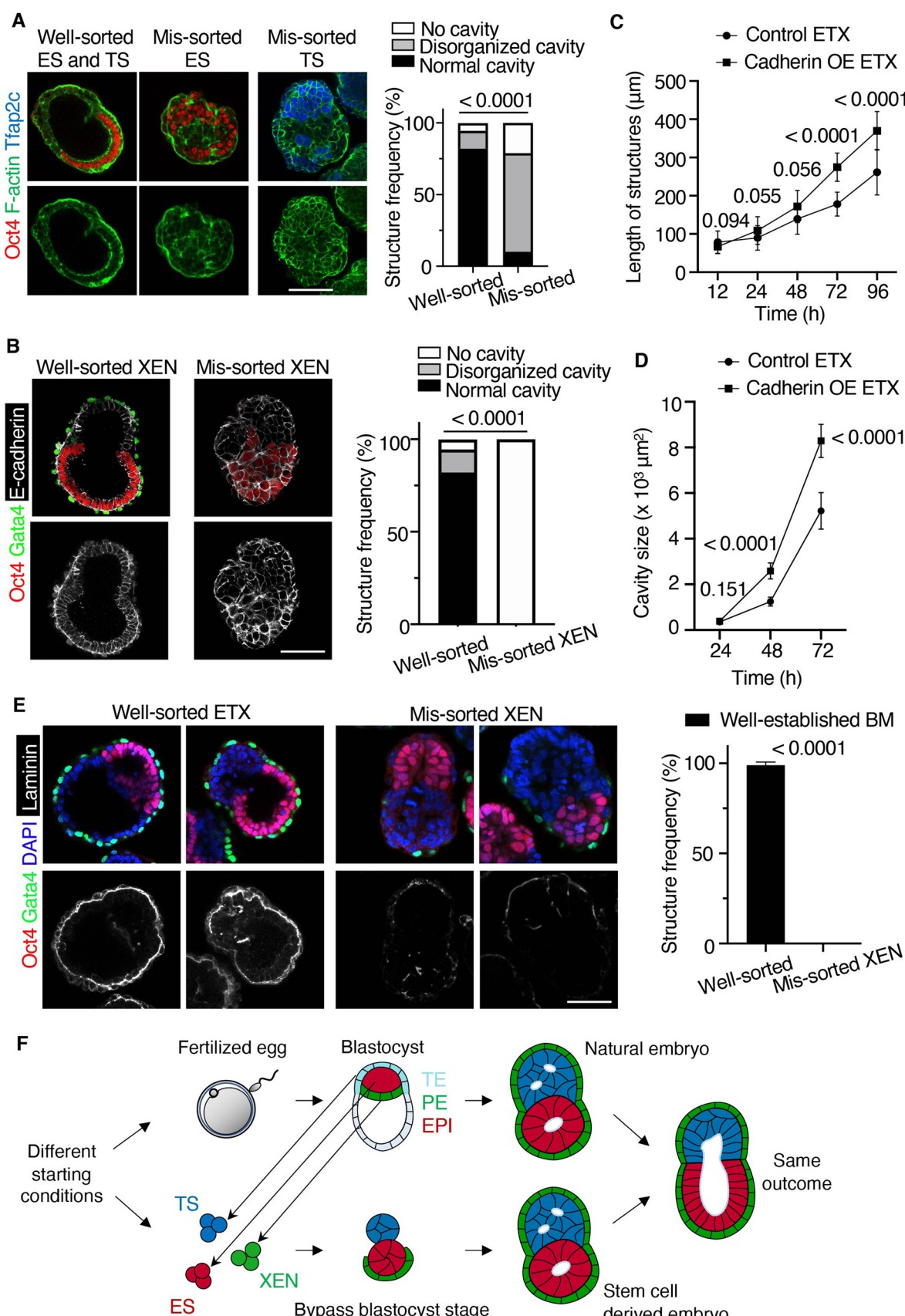

**Extended Data Fig. 6 | See next page for caption.**

**Extended Data Fig. 6 | Correct cell sorting and self-organization is necessary for proper morphogenesis.** (**a**) Comparison and quantification of cavity formation in structures containing mis-sorted ES or TS and (**b**) mis-sorted XEN cells. Well-sorted structures: n = 73; Mis-sorted ES structures: n = 103; Mis-sorted TS structures: n = 109. Mis-sorted XEN structures: n = 57. N = 3 for each condition. Scale bar, 100 μm. Statistics calculated by unpaired two-tailed Student's *t* test. (**c**) The average length and (**d**) internal cavity size of Cadherin OE ETX and control ETX at different time points. 20 to 30 structures were collected at each time points. Data are presented as Mean ± SD. Statistics calculated by unpaired two-tailed Student's *t* test. *P* values indicate significant difference between Cadherin OE and control ETX at the same time point. (**e**) Comparison and quantification of basement membrane formation in structures containing well-sorted and mis-sorted XEN. Well-sorted structures: n = 84; Mis-sorted XEN structures: n = 74. N = 3 for each condition. Data are presented as Mean ± SD. Statistics calculated by unpaired two-tailed Student's *t* test. Scale bar, 100 μm. (**f**) Schematic image shows natural and ETX embryos use different routes to form the post-implantation embryos. In ETX embryos, lineage-specific stem cells bypass the blastocyst structure to directly assemble a post-implantation embryo. Source numerical data are available in source data.

# Reporting Summary

## Statistics

For all statistical analyses, confirm that the following items are present in the figure legend, table legend, main text, or Methods section.

| n/a | Confirmed | |
|---|---|---|
| ☐ | ☒ | The exact sample size (*n*) for each experimental group/condition, given as a discrete number and unit of measurement |
| ☐ | ☒ | A statement on whether measurements were taken from distinct samples or whether the same sample was measured repeatedly |
| ☐ | ☒ | The statistical test(s) used AND whether they are one- or two-sided *Only common tests should be described solely by name; describe more complex techniques in the Methods section.* |
| ☐ | ☒ | A description of all covariates tested |
| ☐ | ☒ | A description of any assumptions or corrections, such as tests of normality and adjustment for multiple comparisons |
| ☐ | ☒ | A full description of the statistical parameters including central tendency (e.g. means) or other basic estimates (e.g. regression coefficient) AND variation (e.g. standard deviation) or associated estimates of uncertainty (e.g. confidence intervals) |
| ☐ | ☒ | For null hypothesis testing, the test statistic (e.g. *F*, *t*, *r*) with confidence intervals, effect sizes, degrees of freedom and *P* value noted *Give P values as exact values whenever suitable.* |
| ☒ | ☐ | For Bayesian analysis, information on the choice of priors and Markov chain Monte Carlo settings |
| ☒ | ☐ | For hierarchical and complex designs, identification of the appropriate level for tests and full reporting of outcomes |
| ☒ | ☐ | Estimates of effect sizes (e.g. Cohen's *d*, Pearson's *r*), indicating how they were calculated |

*Our web collection on statistics for biologists contains articles on many of the points above.*

## Software and code

Policy information about availability of computer code

| | |
|---|---|
| Data collection | LEICA software LAS X was used for image acquisition. |
| Data analysis | For graphical statistics and statistical tests , GraphPad Prism (V8.0 & 7.0a) was used. Fiji image processing software and Imaris software were used for image processing and analysis. JPK IP Software was used for calculating cell adhesion force and cortical stiffness. NanoScope software (V6.13) was used for sample stage calibration before loading the sample during AFM experiment. FlowJo software (V10.7.1) was used to plot E- and P-cadherin intensity, measured from FACS experiment. |

For manuscripts utilizing custom algorithms or software that are central to the research but not yet described in published literature, software must be made available to editors and reviewers. We strongly encourage code deposition in a community repository (e.g. GitHub). See the Nature Portfolio guidelines for submitting code & software for further information.

## Data

Policy information about availability of data

All manuscripts must include a data availability statement. This statement should provide the following information, where applicable:
- Accession codes, unique identifiers, or web links for publicly available datasets
- A description of any restrictions on data availability
- For clinical datasets or third party data, please ensure that the statement adheres to our policy

Previously published scRNA-seq data that were re-analysed here are available under accession code GSE161947. Source numerical data are provided with this paper. All relevant data supporting the key findings of this study are available within the article and its Supplementary Information files. All other data supporting the findings of this study are available from the corresponding author on reasonable request.

# Field-specific reporting

Please select the one below that is the best fit for your research. If you are not sure, read the appropriate sections before making your selection.

☒ Life sciences ☐ Behavioural & social sciences ☐ Ecological, evolutionary & environmental sciences

For a reference copy of the document with all sections, see nature.com/documents/nr-reporting-summary-flat.pdf

# Life sciences study design

All studies must disclose on these points even when the disclosure is negative.

| | |
|---|---|
| Sample size | Sample size was determined based on our previous experience and the work of other groups using stem cell derived embryos and mouse natural embryos as experimental model systems (Nature cell biology 20.8 (2018): 979-989; Developmental cell 56.3 (2021): 366-382) |
| Data exclusions | ETX-embryo samples for scRNA-sequencing analysis: those samples that did not pass the quality controls were excluded from the analysis. |
| Replication | Each result described in the paper is based on at least three independent biological replicates. Figure legends indicate the number of independent experiments performed in each analysis. |
| Randomization | Samples (mouse embryos) were allocated randomly into experimental groups.<br>The in vitro cell experiments were not randomized as it was not necessary.<br>For experiments with chemical inhibitors, samples were randomly allocated to control and experimental groups.<br>Embryos were randomly allocated to control and experimental groups for in vivo experiments. |
| Blinding | The investigators were not blinded to group allocation, because this study investigates the fundamental self-organization principles in both stem cell derived and natural embryos, blinding is not relevant to our study and the experiments were descriptive in their nature. |

# Reporting for specific materials, systems and methods

We require information from authors about some types of materials, experimental systems and methods used in many studies. Here, indicate whether each material, system or method listed is relevant to your study. If you are not sure if a list item applies to your research, read the appropriate section before selecting a response.

## Materials & experimental systems

| n/a | Involved in the study |
|---|---|
| ☐ | ☒ Antibodies |
| ☐ | ☒ Eukaryotic cell lines |
| ☒ | ☐ Palaeontology and archaeology |
| ☐ | ☒ Animals and other organisms |
| ☒ | ☐ Human research participants |
| ☒ | ☐ Clinical data |
| ☒ | ☐ Dual use research of concern |

## Methods

| n/a | Involved in the study |
|---|---|
| ☒ | ☐ ChIP-seq |
| ☐ | ☒ Flow cytometry |
| ☒ | ☐ MRI-based neuroimaging |

# Antibodies

| | |
|---|---|
| Antibodies used | Primary antibodies:<br>Goat polyclonal anti-Tfap2c R&D Systems Cat# AF5059; RRID: AB_2255891 (1:200)<br>Goat polyclonal anti-Brachyury R&D Systems Cat# AF2085; RRID: AB_2200235 (1:200)<br>Rabbit monoclonal anti-Gata4 Cell Signaling Technology Cat# 36966; RRID: AB_2799108 (1:500)<br>Rabbit polyclonal anti-laminin Sigma-Aldrich Cat# L9393; RRID: AB_477163 (1:500)<br>Mouse monoclonal anti-Oct4 Santa Cruz Biotechnology Cat# sc-5279; RRID: AB_628051 (1:500)<br>Rat Monoclonal anti-E-cadherin Thermo Fisher Scientific Cat# 13-1900; RRID: AB_2533005 (1:200)<br>Goat Monoclonal anti-P-cadherin Santa Cruz Biotechnology Cat# sc-1501; RRID: AB_630961 (1:100)<br>Mouse Monoclonal anti-P-cadherin Fisher Scientific Cat# MS-1741; RRID: AB_149083 (1:100)<br><br>Secondary antibodies:<br>Donkey anti-Mouse IgG (H+L), Alexa Fluor 488 Thermo Fisher Scientific Cat# A-21202; RRID: AB_141607 (1:500)<br>Donkey anti-Goat IgG (H+L), Alexa Fluor 488 Thermo Fisher Scientific Cat# A-11055; RRID: AB_2534102 (1:500)<br>Donkey anti-Rat IgG (H+L), Alexa Fluor 488 Thermo Fisher Scientific Cat# A-21208; RRID: AB_2535794 (1:500)<br>Donkey anti-Rabbit IgG (H+L), Alexa Fluor 568 Thermo Fisher Scientific Cat# A10042; RRID: AB_2534017 (1:500)<br>Donkey anti-Mouse IgG (H+L), Alexa Fluor 568 Thermo Fisher Scientific Cat# A10037; RRID: AB_2534013 (1:500)<br>Donkey anti-Rabbit IgG (H+L), Alexa Fluor 647 Thermo Fisher Scientific Cat# A-31573; RRID: AB_2536183 (1:500) |

Donkey anti-Goat IgG (H+L), Alexa Fluor 647 Thermo Fisher Scientific Cat# A-21447; RRID: AB_2535864 (1:500)
Alexa Fluor™ Plus 405 Phalloidin Thermo Fisher Scientific Cat# A30104 (1:200)

| Validation | The subcellular localization of all the proteins analyzed in this study has been previously reported.<br>Tfap2c: It correctly stained TS cells and the extra-embryonic ectoderm cells in post-implantation embryos (Science 356, doi:10.1126/science.aal1810).<br>Brachyury: It correctly stained mesoderm at 6.5 and later as reported and expected (Dev Biol 288, 363-371).<br>Gata4: It correctly labelled XEN cells and the visceral endoderm in postimplantation embryos (eLife 2018;7:e32839)<br>Laminin: It correctly stained the basement membrane between visceral endoderm and Exe or epiblast, as reported elsewhere and as expected (Dev Dyn 241, 270-283)<br>Oct4: It specifically stained ES cells and the epiblast at all stages tested, as expected (Science 356, doi:10.1126/science.aal1810).<br>E-cadherin: It correctly stained the cell-cell junction and basolateral side of cells in the embryo as reported and as expected (Science 356, doi:10.1126/ science.aal1810)<br>P-cadherin: It correctly stained the cell-cell junction as reported (Cell 123.5 (2005): 917-929)<br>F-actin: it correctly stained the cell membrane and was apically enriched, as expected and reported (Development, 138(2011) 3011-3020). |
|---|---|

# Eukaryotic cell lines

Policy information about cell lines

| Cell line source(s) | Experiments were performed using mouse E14 wild-type ES cells (derived in Zernicka-Goetz's lab). Wild-type TS cells (a gift from Jenny Nichols), and wild-type XEN cells (a gift from Ellen Na). Cdh1 and Cdh6 overexpressing ES cells were generated from E14 wild-type ES cells; Cdh3 overexpressing TS cells were generated from wild-type TS cells; Cdh1 and Cdh6 overexpressing XEN cells were generated from wild-type XEN cells (see below). Details can be found in method section in the paper. |
|---|---|
| Authentication | Cells were maintained in conditions to preserve stem cell character and prevent differentiation. Plates were inspected for morphological evidence of differentiation (altered colony morphology in ESC cultures or presence of trophoblast giant cells in TSC cultures..etc) and plates with differentiated cells were discarded. Furthermore, cell identities were confirmed routinely by immunoflourescence marker expressions. |
| Mycoplasma contamination | Cell lines were routinely tested for mycoplasma contamination by PCR |
| Commonly misidentified lines<br>(See ICLAC register) | The cells we used are not part of this database |

# Animals and other organisms

Policy information about studies involving animals; ARRIVE guidelines recommended for reporting animal research

| Laboratory animals | Mice (Mus musculus) were used to obtain mouse embryos for this study. Six-week-old female CD-1 mice (both male and female) were used. All experimental mice were free of pathogens and were on a 12-12 hour light-dark cycle, with unlimited access to water and food. Temperature in the facility was controlled and maintained at 21 oC. |
|---|---|
| Wild animals | The study did not involve wild animals. |
| Field-collected samples | The study did not involve samples collected from the field. |
| Ethics oversight | All experiments involving mice have been regulated by the Animals (Scientific Procedures) Act 1986 Amendment Regulations 2012 and additional ethical review by the University of Cambridge Animal Welfare and Ethical Review Body (AWERB). Experiments were authorised by the Home Office (Licence number: 70/8864). |

Note that full information on the approval of the study protocol must also be provided in the manuscript.

# Flow Cytometry

## Plots

Confirm that:

☒ The axis labels state the marker and fluorochrome used (e.g. CD4-FITC).

☒ The axis scales are clearly visible. Include numbers along axes only for bottom left plot of group (a 'group' is an analysis of identical markers).

☒ All plots are contour plots with outliers or pseudocolor plots.

☒ A numerical value for number of cells or percentage (with statistics) is provided.

## Methodology

| Sample preparation | As described in the Method, ES cells cultured on gelatin-coated plates were trypsinized to generate single cells. TS cells cultured on MEFs were trypsinized into single cells and then plated onto gelatin coated plates for 30 min to eliminate MEFs. |
|---|---|

Single ES and TS cell suspensions were collected, fixed in 4% PFA, and permeabilized for 30 min at room temperature using 0.3% Triton-X-100 and 0.1% glycine. Cells were then incubated with anti E- or P-cadherin antibody for overnight incubation at 4°C in blocking buffer (PBST containing 10% FBS). Cells were washed twice in PBST and then incubated with secondary antibody (1:500 dilution) in blocking buffer at room temperature for 1-2h.

Instrument

flow cytometry (BD Biosciences)

Software

FlowJo software (https://www.flowjo.com)

Cell population abundance

We performed FACS experiment to identify the intensity distribution of E- and P-cadherin, no cell population were sorted in this work

Gating strategy

1.) SSC vs. FSC gating to exclude debris. 2.) FSC-H vs. FSC-A gating to exclude doublets.

☒ Tick this box to confirm that a figure exemplifying the gating strategy is provided in the Supplementary Information.

