## [Peer Review File · Nature Cell Biology]

Peer Review Information

Journal: Nature Cell Biology

Manuscript Title: Stem cell-derived synthetic embryos self-assemble by exploiting cadherin codes and cortical tension

Corresponding author name(s): Magdalena Zernicka-Goetz

Reviewer Comments & Decisions:

Decision Letter, initial version:

*Please delete the link to your author homepage if you wish to forward this email to co-authors.

Dear Professor Zernicka-Goetz,

Your manuscript, "Stem cell-derived embryos self-assemble by repurposing pre- and post-implantation cadherin codes", has now been seen by 3 referees, who are experts in engineered embryo models and mammalian development (referee 1); early mammalian development and embryonic assembly (referee 2); and cadherins, adhesion and development (referee 3). As you will see from their comments (attached below) they overall find this work of potential interest, but have raised some concerns (referee 3, in particular), which in our view would need to be addressed with considerable revisions before we can consider publication in Nature Cell Biology.

Nature Cell Biology editors discuss the referee reports in detail within the editorial team, including the chief editor, to identify key referee points that should be addressed with priority. To guide the scope of the revisions, I have listed these points below. We are committed to providing a fair and constructive peer-review process, so please feel free to contact me if you would like to discuss any of the referee comments further.

In particular, it would be essential to:

(A) Cite and contextualize previous literature suggested by referee 3.

(B) Provide a more robust statistical analysis and clearer presentation of the data, per referee 3.

(C) All other referee comments pertaining to strengthening existing data, additional analyses, methodological details, clarifications and textual changes should also be addressed.

(D) Finally please pay close attention to our guidelines on statistical and methodological reporting (listed below) as failure to do so may delay the reconsideration of the revised manuscript. In particular please provide:

- a Supplementary Table including all numerical source data in Excel format, with data for different figures provided as different sheets within a single Excel file. The file should include source data giving rise to graphical representations and statistical descriptions in the paper and for all instances where the figures present representative experiments of multiple independent repeats, the source

data of all repeats should be provided.

We would be happy to consider a revised manuscript that would satisfactorily address these points, unless a similar paper is published elsewhere, or is accepted for publication in Nature Cell Biology in the meantime.

- ensure that it conforms to our format instructions and publication policies (see below and www.nature.com/nature/authors/).
- provide a point-by-point rebuttal to the full referee reports verbatim, as provided at the end of this letter.
- provide the completed Editorial Policy Checklist (found here <https://www.nature.com/authors/policies/Policy.pdf>), and Reporting Summary (found here <https://www.nature.com/authors/policies/ReportingSummary.pdf>). This is essential for reconsideration of the manuscript and these documents will be available to editors and referees in the event of peer review. For more information see <http://www.nature.com/authors/policies/availability.html> or contact me.

Nature Cell Biology is committed to improving transparency in authorship. As part of our efforts in this direction, we are now requesting that all authors identified as 'corresponding author' on published papers create and link their Open Researcher and Contributor Identifier (ORCID) with their account on the Manuscript Tracking System (MTS), prior to acceptance. ORCID helps the scientific community achieve unambiguous attribution of all scholarly contributions. You can create and link your ORCID from the home page of the MTS by clicking on 'Modify my Springer Nature account'. For more information please visit www.springernature.com/orcid.

[REDACTED]

We would like to receive a revised submission within six months. We would be happy to consider a revision even after this timeframe, however if the resubmission deadline is missed and the paper is eventually published, the submission date will be the date when the revised manuscript was received.

We hope that you will find our referees' comments, and editorial guidance helpful. Please do not hesitate to contact me if there is anything you would like to discuss.

Best wishes,

Stelios

Stylios Lefkopoulos, PhD
He/him/his
Associate Editor
Nature Cell Biology
Springer Nature
Heidelberger Platz 3, 14197 Berlin, Germany

E-mail: stylianos.lefkopoulos@springernature.com
Twitter: @s_lefkopoulos

Reviewers' Comments:

Reviewer #1:

Remarks to the Author:

The manuscript "Stem cell-derived embryos self-assemble by repurposing pre- and post-implantation cadherin codes" is a very nicely executed and elegant work from the Zernicka-Goetz group, providing clarification on the role of differential cell adhesion, due the expression of different Cadherins (Cdh1, Cdh3, Cdh6), to the optimal sorting of cellular compartments to generate EPI-like, VE-like and TE-like compartments that closely mimic the in vivo cell-sorting.

The authors show nicely that differential expression of 3 cadherins help to trigger cellular compartmentalization, luminogenesis and formation of the basement membrane. This resulted in an optimization of their method, resulting triple the number of properly formed ETX-embryos.

The study is of very high quality and robust, nicely presented, with adequate statistics and a crystal clear message.

My only and major concern is whether the message presented is strong and novel enough, providing enough conceptual advances to the field.

Reviewer #2:

Remarks to the Author:

In this paper, Bao et al demonstrate that a cadherin code regulates the assembly and sorting of the first three cell lineages during mammalian development using a stem cell model recapitulating this process in vitro (ETX embryos). Recent work from many labs has been intensely focus on elucidating mechanism to assembly synthetic embryos.

This paper represents an important contribution to the field, as it shows that some of the mechanisms are conserved between the two systems. The findings are important and should encourage future studies to also test to which extent their in vitro protocols for assembling embryoids and organoids follow a similar sequence of events, compared to the in vivo systems.

The identification of a cadherin code and a link to transcriptional regulation will be of interest to stem cell and developmental biologists, as well as to the mechanobiology field. The paper combines various genetic and biophysical methods and the data support the main conclusions.

Reviewer #3:

Remarks to the Author:

In this manuscript, Bao et al. explore the role of adhesion-based sorting in the self-organization of ETX-embryos, an emerging model system for mouse embryogenesis. The authors use an impressive array of state-of-the-art techniques, including AFM and imaging-based force inference, alongside perturbations to cadherin expression, in order to very nicely and directly demonstrate that a combinatorial cadherin code drives cell type sorting in the ETX-embryo. Further, by optimizing expression levels of the cadherin code molecules, the authors are able to greatly increase ETX-embryo sorting efficiency.

Adhesion-based sorting as a mechanism to pattern cells is half century-old idea that has been well-

established *in vitro*, *in vivo*, and *in silico*. Combinatorial adhesion codes, such as the one discovered in this work, hold the potential to explain increasingly complex developmental patterns, and represent a new frontier for this well-established field. To my knowledge this is the only work aside from a recent Science paper (<https://www.science.org/doi/10.1126/science.aba6637>) which directly demonstrates (using force spectroscopy) a combinatorial adhesion code in a developing organism. Additionally, cell derived embryos represent an exciting new model system with clear human health applications, as well as implications more broadly for organoid biology and bioengineering. Their insight into how to robustly design ETX-embryos stands to be very impactful for a broad field of research. The work will be of great interest to molecular, cell, and developmental biologists, as well as biophysicists and bioengineers, and therefore should be appropriate for the broad readership of Nature Cell Biology.

Major comments

- Many previous works have identified cortical tension as another common driver of embryonic cell sorting (e.g., Krieg et al., NCB, 2008). In particular, Takashi Hiiragi's lab has outlined such roles for cortical tension in mouse embryos at similar developmental stages (e.g., Niwayama et al. 2019, <https://doi.org/10.1016/j.devcel.2019.10.012>). This previous literature also delineated the effects of cortical tension on lumenization, another major topic of this work. The authors should cite and contextualize their results in the context of this literature. Additionally, to rule out the possibility that differences in cortical tension are contributing to sorting in ETX-embryos, the authors should perform AFM measurements of cortical tension, ideally including a subset of cadherin OE and KD conditions.

- The authors' interpretation that a cadherin code mediates sorting in ETX embryos is largely dependent on the sorting deficiencies that arise from cadherin OE and KD, presented in Figure 3. However, these data are difficult to interpret. The use of pie charts to describe sorting outcomes makes it difficult to discern whether the effects are statistically significant. Further, a pie chart implies all outcomes are mutually exclusive, but it is unclear to me why multiple compartments cannot be mis-sorted at once. Finally, the pie slices do not add up to 100%, and neither do the stacked bar charts. The authors should provide a more robust statistical analysis and clearer presentation of the data.

- The present manuscript largely ignores a broad swath of foundational literature on adhesion-mediated sorting, in particular those covering the differential adhesion hypothesis / differential interfacial tension hypothesis (DAH/DITH) (e.g., Brodland et al., 2002 <https://doi.org/10.1115/1.1449491> and Canty et al., 2017 <https://doi.org/10.1038/s41467-017-00146-x>, but there are many others). These classic works provide clear expectations for how differential cell-cell adhesion, and thus differential interfacial tension, drives sorting of various patterns. For this system - given the relative homotypic tensions $T_{\rightarrow ES-ES}$, $T_{\rightarrow TS-TS}$, $T_{\rightarrow XEN-XEN}$, and heterotypic tensions $T_{\rightarrow ES-TS}$, $T_{\rightarrow ES-XEN}$, and $T_{\rightarrow TS-XEN}$, one can determine what the final sorted pattern should be. The inverse is also true: given a sorted pattern, one can infer the relative tensions/adhesion strengths (e.g., for Day 3 ETX-embryos, one would expect $(T_{\rightarrow ES-ES} \sim T_{\rightarrow TS-TS}) < T_{\rightarrow ES-TS} < (T_{\rightarrow ES-XEN} \sim T_{\rightarrow TS-XEN}) < T_{\rightarrow XEN-XEN}$). The authors should cite and contextualize their results in the context of this literature, and make it clear how their results would be interpreted using this existing formalism.

In particular I believe there is a discrepancy between the predicted interfacial hierarchy for the sorted configuration in the ETX-embryo [$(T_{\rightarrow ES-ES} \sim T_{\rightarrow TS-TS}) < T_{\rightarrow ES-TS} < (T_{\rightarrow ES-XEN} \sim T_{\rightarrow TS-XEN}) < T_{\rightarrow XEN-XEN}$], and the measured ones (From Figure 2C: $(T_{\rightarrow ES-ES} \sim T_{\rightarrow TS-TS}) < T_{\rightarrow ES-XEN} < (T_{\rightarrow XEN-XEN} \sim T_{\rightarrow ES-TS} \sim T_{\rightarrow TS-XEN})$). The measured forces should lead to a structure where XEN cells form a closed layer around the ES cells that occludes the ES-TS cell contacts (because $T_{\rightarrow ES-XEN} < T_{\rightarrow ES-TS}$). The XEN cells are not predicted to envelop the TS layer (because $T_{\rightarrow XEN-XEN} \sim T_{\rightarrow TS-XEN}$). Please explain this discrepancy.

Minor comments

- In the context of sorting deficiencies in cadherin KD and OE conditions, is sorting completely inhibited, or just slower? I ask because the answer determines whether I interpret the results as the cadherin code being the sole determinant of sorting, or whether it is simply an accessory mechanism.

- On a related note, in the cadherin OE cases that improve sorting efficiency, it would be appropriate to cite theory papers showing that larger quantitative differences in cadherin expression between distinct cell types increases the rate of sorting (<https://doi.org/10.1371/journal.pone.0024999>). Again, I wonder whether it is the rate of sorting or the robustness of sorting that is increased in Figure 3H. Performing time course measurements of sorting efficiency may address this question.
- The authors mention that the broad range of measured cohesion forces within a population could explain why such a large fraction of the EXT-embryos exhibit mis-sorted configurations. One way to test this hypothesis would be to use bootstrapping of the interaction force measurements to determine what fraction of embryos, whose cells have interaction forces randomly sampled from the measured distribution, are predicted to have the proper hierarchy to produce the correct sorted pattern.
- Cdh3 OE ES cells localizing to the periphery of the TE layer (Figure 1E-F) can be explained by $T \rightarrow TE-TE < T \rightarrow ES(OCECdh3)-TS < T \rightarrow ES(OCECdh3)-ES(OCECdh3)$, according to the differential adhesion hypothesis. It implies the OE of Cdh3 allows strong enough binding to TE cells to overcome the native ES-ES interactions mediated by Cdh1.
- In the main text, the figure reference for the contact angle force inference (Figure 2D) is mislabeled as Figure 1D.
- Figure 1
 - o PE and VE seem to be used interchangeably. The term VE is used in Figure 1D, but the same layer is labeled PE in Figure 1A. It might be helpful to use a consistent terminology.
- Figure 2
 - o You might label the y-axis of G "Homotypic cohesion force" for clarity.
 - o For Figure 2E, it would be nice to have statistical significance for all comparisons represented.
 - o Will you please discuss in the text why KD of either Cdh1 or CDH3 is sufficient to abrogate homotypic adhesion of TS cells? I would expect that both would have to be removed to eliminate adhesion.

AUTHOR AFFILIATIONS – should be denoted with numerical superscripts (not symbols) preceding

the names. Full addresses should be included, with US states in full and providing zip/post codes. The corresponding author is denoted by: "Correspondence should be addressed to [initials]."

Methods should be written concisely, but should contain all elements necessary to allow interpretation and replication of the results. As a guideline, Methods sections typically do not exceed 3,000 words. The Methods should be divided into subsections listing reagents and techniques. When citing previous methods, accurate references should be provided and any alterations should be noted. Information must be provided about: antibody dilutions, company names, catalogue numbers and clone numbers for monoclonal antibodies; sequences of RNAi and cDNA probes/primers or company names and catalogue numbers if reagents are commercial; cell line names, sources and information on cell line identity and authentication. Animal studies and experiments involving human subjects must be reported in detail, identifying the committees approving the protocols. For studies involving human subjects/samples, a statement must be included confirming that informed consent was obtained. Statistical analyses and information on the reproducibility of experimental results should be provided in a section titled "Statistics and Reproducibility".

All Nature Cell Biology manuscripts submitted on or after March 21 2016 must include a Data availability statement at the end of the Methods section. For Springer Nature policies on data

availability see <http://www.nature.com/authors/policies/availability.html>; for more information on this particular policy see <http://www.nature.com/authors/policies/data/data-availability-statements-data-citations.pdf>. The Data availability statement should include:

- Accession codes for primary datasets (generated during the study under consideration and designated as "primary accessions") and secondary datasets (published datasets reanalysed during the study under consideration, designated as "referenced accessions"). For primary accessions data should be made public to coincide with publication of the manuscript. A list of data types for which submission to community-endorsed public repositories is mandated (including sequence, structure, microarray, deep sequencing data) can be found here <http://www.nature.com/authors/policies/availability.html#data>.
- Unique identifiers (accession codes, DOIs or other unique persistent identifier) and hyperlinks for datasets deposited in an approved repository, but for which data deposition is not mandated (see here for details <http://www.nature.com/sdata/data-policies/repositories>).
- At a minimum, please include a statement confirming that all relevant data are available from the authors, and/or are included with the manuscript (e.g. as source data or supplementary information), listing which data are included (e.g. by figure panels and data types) and mentioning any restrictions on availability.
- If a dataset has a Digital Object Identifier (DOI) as its unique identifier, we strongly encourage including this in the Reference list and citing the dataset in the Methods.

We recommend that you upload the step-by-step protocols used in this manuscript to the Protocol Exchange. More details can be found at www.nature.com/protocolexchange/about.

All imaging data should be accompanied by scale bars, which should be defined in the legend. Cropped images of gels/blots are acceptable, but need to be accompanied by size markers, and to retain visible background signal within the linear range (i.e. should not be saturated). The boundaries of panels with low background have to be demarked with black lines. Splicing of panels should only be considered if unavoidable, and must be clearly marked on the figure, and noted in the legend with a statement on whether the samples were obtained and processed simultaneously. Quantitative comparisons between samples on different gels/blots are discouraged; if this is unavoidable, it should only be performed for samples derived from the same experiment with gels/blots were processed in parallel, which needs to be stated in the legend.

- For line art, graphs, charts and schematics we prefer Adobe Illustrator (.AI), Encapsulated PostScript (.EPS) or Portable Document Format (.PDF). Files should be saved or exported as such directly from the application in which they were made, to allow us to restyle them according to our journal house style.
- We accept PowerPoint (.PPT) files if they are fully editable. However, please refrain from adding PowerPoint graphical effects to objects, as this results in them outputting poor quality raster art. Text used for PowerPoint figures should be Helvetica (preferred) or Arial.
- We do not recommend using Adobe Photoshop for designing figures, but we can accept Photoshop generated (.PSD or .TIFF) files only if each element included in the figure (text, labels, pictures, graphs, arrows and scale bars) are on separate layers. All text should be editable in 'type layers' and line-art such as graphs and other simple schematics should be preserved and embedded within 'vector smart objects' - not flattened raster/bitmap graphics.
- Some programs can generate Postscript by 'printing to file' (found in the Print dialogue). If using an application not listed above, save the file in PostScript format or email our Art Editor, Allen Beattie for advice (a.beattie@nature.com).

The total number of Supplementary Figures (not including the “unprocessed scans” Supplementary Figure) should not exceed the number of main display items (figures and/or tables (see our Guide to Authors and March 2012 editorial

<http://www.nature.com/ncb/authors/submit/index.html#suppinfo>;

<http://www.nature.com/ncb/journal/v14/n3/index.html#ed>). No restrictions apply to

Supplementary Tables or Videos, but we advise authors to be selective in including supplemental data.

GUIDELINES FOR EXPERIMENTAL AND STATISTICAL REPORTING

REPORTING REQUIREMENTS – To improve the quality of methods and statistics reporting in our papers we have recently revised the reporting checklist we introduced in 2013. We are now asking all life sciences authors to complete two items: an Editorial Policy Checklist (found here <https://www.nature.com/authors/policies/Policy.pdf>) that verifies compliance with all required editorial policies and a reporting summary (found here <https://www.nature.com/authors/policies/ReportingSummary.pdf>) that collects information on experimental design and reagents. These documents are available to referees to aid the evaluation of the manuscript. Please note that these forms are dynamic ‘smart pdfs’ and must therefore be downloaded and completed in Adobe Reader. We will then flatten them for ease of use by the reviewers. If you would like to reference the guidance text as you complete the template, please access these flattened versions at <http://www.nature.com/authors/policies/availability.html>.

We strongly recommend the presentation of source data for graphical and statistical analyses as a separate Supplementary Table, and request that source data for all independent repeats are provided when representative experiments of multiple independent repeats, or averages of two

independent experiments are presented. This supplementary table should be in Excel format, with data for different figures provided as different sheets within a single Excel file. It should be labelled and numbered as one of the supplementary tables, titled "Statistics Source Data", and mentioned in all relevant figure legends.

Author Rebuttal to Initial comments

Reviewers' Comments (in blue) and our Responses (in black):

Reviewer #1:

Remarks to the Author:

The manuscript "Stem cell-derived embryos self-assemble by repurposing pre- and post-implantation cadherin codes" is a very nicely executed and elegant work from the Zernicka-Goetz group, providing clarification on the role of differential cell adhesion, due the expression of different Cadherins (Cdh1, Cdh3, Cdh6), to the optimal sorting of cellular compartments to generate EPI-like, VE-like and TE-like compartments that closely mimic the in vivo cell-sorting.

The authors show nicely that differential expression of 3 cadherins help to trigger cellular compartmentalization, luminogenesis and formation of the basement membrane. This resulted in an optimization of their method, resulting triple the number of properly formed ETX-embryos. The study is of very high quality and robust, nicely presented, with adequate statistics and a cristal clear message.

My only and major concern is whether the message presented is strong and novel enough, providing enough conceptual advances to the field.

Response: We very much thank the referee for these supportive comments.

We believe that the message we present provides a novel and strong conceptual advance to the field. Before our study, it remained unknown how multiple stem cell types are able to self-organize into a functional and physiologically relevant embryo-like structure, a synthetic embryo. Our work shows for the first time that the assembly of synthetic embryos from three stem cell types occurs through a distinct sequence of events compared to that of natural embryos – in that it skips the pre-implantation stage of the blastocyst – but it still uses the physiologically relevant differential expression of cadherins observed between cell types in natural development. Despite natural and synthetic embryos arising through very different events (fertilization vs the random mixing of stem cells in a dish), the process and outcomes of development are similar. Through several lines of independent experimentation, we provide evidence that disrupting these assembly instructions (particularly the expression of specific cadherins, the so-called "cadherin code") compromises synthetic embryo development. Importantly, we capitalize upon the cadherin code to dramatically improve the efficiency of synthetic embryo formation (by 3-fold) and thus substantially enhance the utility of this synthetic embryo model. These results provide an important contribution to the field as they show that several of the assembly mechanisms are conserved between natural and synthetic systems. The findings will encourage future studies to test the extent to which in vitro protocols for assembling embryoids and organoids follow a similar sequence of events, compared to the in vivo systems.

We are confident that our findings will have general relevance in understanding mechanisms that establish tissue architecture and will be of great interest to molecular, cell, stem cell and developmental biologists, as well as biophysicists and bioengineers, and therefore for the broad readership of Nature Cell Biology, as actually other referees note themselves.

Reviewer #2:

Remarks to the Author:

In this paper, Bao et al demonstrate that a cadherin code regulates the assembly and sorting of the first three cell lineages during mammalian development using a stem cell model recapitulating this process in vitro (ETX embryos). Recent work from many labs has been intensely focus on elucidating mechanism to assembly synthetic embryos. This paper represents an important contribution to the field, as it shows that some of the

mechanisms are conserved between the two systems. The findings are important and should encourage future studies to also test to which extent their *in vitro* protocols for assembling embryoids and organoids follow a similar sequence of events, compared to the *in vivo* systems.

The identification of a cadherin code and a link to transcriptional regulation will be of interest to stem cell and developmental biologists, as well as to the mechanobiology field. The paper combines various genetic and biophysical methods and the data support the main conclusions.

Response: We very much thank the reviewer for their supportive comments.

Reviewer #3:

Remarks to the Author:

In this manuscript, Bao et al. explore the role of adhesion-based sorting in the self-organization of ETX-embryos, an emerging model system for mouse embryogenesis. The authors use an impressive array of state-of-the-art techniques, including AFM and imaging-based force inference, alongside perturbations to cadherin expression, in order to very nicely and directly demonstrate that a combinatorial cadherin code drives cell type sorting in the ETXembryo. Further, by optimizing expression levels of the cadherin code molecules, the authors are able to greatly increase ETX-embryo sorting efficiency.

Adhesion-based sorting as a mechanism to pattern cells is half century-old idea that has been well-established *in vitro*, *in vivo*, and *in silico*. Combinatorial adhesion codes, such as the one discovered in this work, hold the potential to explain increasingly complex developmental patterns, and represent a new frontier for this well-established field.

To my knowledge this is the only work aside from a recent Science paper (<https://www.science.org/doi/10.1126/science.aba6637>) which directly demonstrates (using force spectroscopy) a combinatorial adhesion code in a developing organism. Additionally, cell derived embryos represent an exciting new model system with clear human health applications, as well as implications more broadly for organoid biology and bioengineering. Their insight into how to robustly design ETX-embryos stands to be very impactful for a broad field of research. The work will be of great interest to molecular, cell, and developmental biologists, as well as biophysicists and bioengineers, and therefore should be appropriate for the broad readership of Nature Cell Biology.

Major comments:

- Many previous works have identified cortical tension as another common driver of embryonic cell sorting (e.g., Krieg et al., NCB, 2008). In particular, Takashi Hiiragi's lab has outlined such roles for cortical tension in mouse embryos at similar developmental stages (e.g., Niwayama et al. 2019, <https://doi.org/10.1016/j.devcel.2019.10.012>). This previous literature also delineated the effects of cortical tension on lumenization, another major topic of this work. The authors should cite and contextualize their results in the context of this literature.

Response: We thank the reviewer for these comments. We are aware of the interesting studies of the roles of cortical tension from Hiiragi's lab and didn't relate to them here because their work concern events in preimplantation embryos, in which tension plays a different role in spindle orientation and establishing the blastocyst that becomes a fluid filled ball. The ETX embryo system we describe here, assembles into a structure resembling a post-implantation egg cylinder, whose amniotic cavity assembles through a different route as its cells are arranged with the opposite polarity from those of the blastocyst.

That said, we now have specifically related our studies to the literature about the importance of cell tension, as requested, as indeed tension does have a role alongside cell adhesion in ETX embryo assembly, particularly in relation to the fixation of compartments and externalization of XEN cells.

As requested, we place our findings in the context of past literature and now insert the following paragraph in the discussion part of the revised manuscript:

"The outcome of cell sorting has been previously modelled by considering cell-specific differences in interfacial energies that maximize the most energetically favorable cell interfaces (*Graner et al., 2013; Foty et al., 2015; Steinberg et al., 1963; Cerchiari et al., 2015; Yanagida et al., 2022*). Disparity in interfacial energy was considered to reflect differences in adhesion with cadherins being the best characterized molecular effectors (Nose et al., 1988; Tsai et al., 2020), as espoused in the differential adhesion hypothesis (DAH) (*Foty et al., 2015; Steinberg et al., 1970*). In accord with this hypothesis, we now show cell sorting is driven in ETX-embryos

by the increased strength of cadherin-mediated homotypic interactions in relation to heterotypic interactions. The later development of the differential interfacial tension hypothesis (DITH) (Amack *et al.*, 2012; Brodland *et al.*, 2002; Niwayama *et al.*, 2019; Canty *et al.*, 2017; Krieg *et al.*, 2008) invoking the role of differential cortical tension in sorting has resonance with our findings on XEN cell externalization in the self-assembly process. Together, our observations support this balance between adhesion and tension (DAH vs DITH) as in classical biophysical models of cell sorting. However, incomplete ES-TS sorting still results in “local” order, emphasizing a need for “global” scale sorting to fully recapitulate natural morphogenesis. Indeed, DAH and DITH only account for local sorting to form homotypic clusters of ES and TS cells as seen even in mis-sorted structures. For complete sorting, ETX-embryos must escape from locally correct neighborhoods within globally incorrect patterns to explore alternative conformations. If cells remain in “local minima” before cell-sorting is complete, structures will remain mis-sorted.”

Additionally, to rule out the possibility that differences in cortical tension are contributing to sorting in ETX-embryos, the authors should perform AFM measurements of cortical tension, ideally including a subset of cadherin OE and KD conditions.

Response: We thank the referee for this suggestion. Although our study is focused on the role of differential adhesion in self-organization of ETX embryos, we agree that cortical tension is also important for this process. We have therefore performed the following additional experiments and included them in the revised manuscript (page 6, Figure 3G-I):

“Previous studies have demonstrated a role for cortical stiffness in cell sorting, particularly in cell externalization (Canty *et al.*, 2017; Krieg *et al.*, 2008; Palsson, 2008), prompting us to consider whether cortical tension may influence the capacity of XEN-cells to form their external monolayer. Indeed, our AFM measurements indicated that cortical stiffness is lower in XEN-cells than in either TS- or ES-cells (Figure R1A). To determine whether differences in cortical stiffness between the different stem cell types of ETX-embryos were due to differential actomyosin activity, as in other systems (Harris and Tepass, 2010; Krieg *et al.*, 2008; Salbreux *et al.*, 2012), we measured cortical stiffness in the presence of blebbistatin, a specific inhibitor of myosin II activity (Kovács *et al.*, 2004). Blebbistatin reduced the cortical stiffness of both ES- and TS-cells to the same level as XEN-cells (Figure R1B). We also found that well-sorted ETX embryos at day 3 treated with either blebbistatin or cytochalasin D (actin depolymerizer) for 24h, once the primary sorting phase was completed, failed to maintain sorting efficiently in comparison to control ETX-embryos (Figure R1B). Moreover, when we treated well-sorted ETX-embryos with either blebbistatin or cytochalasin D for 24h, at day 3 once the primary cell sorting phase was complete, more than 80% and 85% of blebbistatin- and cytochalasin D-treated structures respectively, failed to maintain sorting in comparison to 18% of control ETX-embryos (Figure R1B).

To further test the role of cortical stiffness on XEN-cell externalization, we utilized Cellular Potts Model in which cortical stiffness can be independently tuned. When we varied cortical stiffness of XEN-cells in silico, we found that lower stiffness increased both the sorting efficiency and speed of XEN-cell externalization (Figure R1C), suggesting that the softness of XEN-cells is important for this event. Together, these data suggest that in addition to differential expression of distinct cadherins, cortical stiffness also plays an important role in self-assembly of stem cells into ETX-embryos.

Figure R1. (A) Cortical stiffness measurements for indicated cell types before and after treatment with Blebbistatin (Bleb). 30 to 50 cells were measured for each condition. NS, no significant difference; * $p < 0.05$ and ** $p < 0.01$ indicate significance. (B) Day 3 well-sorted ETX embryos were cultured with either blebbistatin, cytochalasin D or DMSO (control) during consolidation stage for 24 hrs and immuno-stained to reveal the indicated markers. Gata4 (green) marks XEN cells; Oct4 (red) marks ES cells; and Tfp2c (blue) marks TS cells. Quantification shows the percentage of disorganized ETX structures. Scale bar represents 100 μm . (C) CPM modelling figure shows the effect of XEN cell stiffness on externalization efficiency. XEN externalization was

quantified by the average radial polarity over 15 simulations for each parameter choice. The sorting efficiency calculated for each time-point is plotted as a heat-map. Time (MCS) indicates the number of stimulation steps.

- The authors' interpretation that a cadherin code mediates sorting in ETX embryos is largely dependent on the sorting deficiencies that arise from cadherin OE and KD, presented in Figure 3. However, these data are difficult to interpret. The use of pie charts to describe sorting outcomes makes it difficult to discern whether the effects are statistically significant. Further, a pie chart implies all outcomes are mutually exclusive, but it is unclear to me why multiple compartments cannot be mis-sorted at once. Finally, the pie slices do not add up to 100%, and neither do the stacked bar charts. The authors should provide a more robust statistical analysis and clearer presentation of the data.

Response: We thank the referee for this suggestion and now provide a clearer presentation of the data. Firstly, we now divide the sorting outcomes into "well-sorted" and "mis-sorted" and present the data in bar charts (new Figure 3G). Secondly, we divide "mis-sorted" structures into "mis-sorted ES", "mis-sorted TS", "mis-sorted XEN" and "others", and present these data in pie charts (new Figure S4A)

Our pie slices do not add up to 100% because some structures are damaged and lost during staining and processing and are therefore not counted. We have added the category "other" in the figure to include those structures and explain their origins.

- The present manuscript largely ignores a broad swath of foundational literature on adhesion-mediated sorting, in particular those covering the differential adhesion hypothesis / differential interfacial tension hypothesis (DAH/DITH) (e.g., Brodland et al., 2002 <https://doi.org/10.1115/1.1449491> and Canty et al., 2017 <https://doi.org/10.1038/s41467-017-00146-x>, but there are many others). These classic works provide clear expectations for how differential cell-cell adhesion, and thus differential interfacial tension, drives sorting of various patterns. For this system - given the relative homotypic tensions T_{ES-ES} , T_{TS-TS} , $T_{XEN-XEN}$, and heterotypic tensions T_{ES-TS} , T_{ES-XEN} , and T_{TS-XEN} , one can determine what the final sorted pattern should be. The inverse is also true: given a sorted pattern, one can infer the relative tensions/adhesion strengths (e.g., for Day 3 ETX-embryos, one would expect $(T_{ES-ES} \sim T_{TS-TS}) < T_{ES-TS} < (T_{ES-XEN} \sim T_{TS-XEN}) < T_{XEN-XEN}$). The authors should cite and contextualize their results in the context of this literature, and make it clear how their results would be interpreted using this existing formalism.

Response: We thank the referee for this helpful comment. We have addressed this point above, in our response to the earlier question on the relative roles of adhesion and tension and discuss this in the revised manuscript, as suggested. Further, we utilized our Cellular Potts Model simulation framework to test whether measured adhesion forces under AFM are sufficient to explain sorting towards an ETX-conformation instead of the 15 other sorted conformations possible with three cell-types. We bootstrap-sample the AFM measurements on a cell-by-cell basis to parameterize the strengths of adhesion between pairs of cells (Figure R2A). Analysing ensembles of simulations with independent samplings (N=498), we calculate the percentages of each conformation. This demonstrates that the ETX-like conformation is the most abundant, followed by conformations where XEN cells envelop the ES compartment (Figure R2B).

We now present this simulation in Figure 2H and I.

Figure R2. (A) Heatmap of adhesion parameter matrix, generated by sampling measured AFM adhesion forces, which parameterizes the CPM. (B) Bootstrapping procedure to infer distributions of conformations given adhesion measurements. Inferred distributions of conformations over time under the CPM (N=498). Schematics represent all possible sorted conformations, demonstrating the ETX-like configuration is most represented. Conformations observed at a frequency of less than 5% are grouped.

In particular I believe there is a discrepancy between the predicted interfacial hierarchy for the sorted configuration in the ETX-embryo [(T-ES-ES ~ T-TS-TS) < T-ES-TS < (T-ES-XEN ~ T-TS-XEN) < T-XENXEN], and the measured ones (From Figure 2C: (T-ES-ES ~ T-TS-TS) < T-ES-XEN < (T-XEN-XEN ~ T-ES-TS ~ T-TS-XEN)). The measured forces should lead to a structure where XEN cells form a closed layer around the ES cells that occludes the ES-TS cell contacts (because T-ES-XEN < T-ES-TS). The XEN cells are not predicted to envelop the TS layer (because T-XEN-XEN ~ T-TS-XEN). Please explain this discrepancy.

Response: We thank the referee for this excellent point, which we have now clarified in the revised manuscript. Indeed, in CPM simulations we observe the conformation the referee predicts in ~10% of samples. The discrepancy the referee highlights is due to the low number of XEN cells we used for making ETX embryos. The optimized number of seeding cells for making ETX embryos is 4~5 XEN cells, 6 ES cells and 12~15 TS cells. With this low number of XEN cells, there are insufficient XEN cells to cover all ES cells during the time available for sorting (Figure R3A) and this allows TS cells to attach to ES cells to form stable adhesions. During the sorting phase, the XEN cells first cover the ES cells and then spread onto the TS compartment to cover the entire structure (Figure R3A). When we increased the number of seeding XEN cells to 10, the XEN cells completely cover the ES cells and consequently prevent TS cells from attaching to ES cells (Figure R3A and B), consistent with our measurements of differential adhesion.

We have added these data into the manuscript as Figure S3E and F.

Figure R3. (A) Examples of structures made from low (control) and high number of XEN cells, stained at day-1 and day-3 to reveal ES cells (Oct4, red), TS cells (Tfap2c, blue) and XEN cells (Gata4, green). Scale bar represents 10 μ m. (B) Schematic of morphological transitions when using low and high number of XEN cells. When XEN cell number is low, XEN cells first attach to ES cells, and ES and TS cells forming apposing compartments, XEN cells spread on TS cells to cover the whole structure. When XEN cell number is high, XEN cells cover ES cells after sorting, and prevent TS cells from attaching to ES cells.

Minor comments

- In the context of sorting deficiencies in cadherin KD and OE conditions, is sorting completely inhibited, or just slower? I ask because the answer determines whether I interpret the results as the cadherin code being the sole determinant of sorting, or whether it is simply an accessory mechanism.

Response: We see adhesion being a major determinant of sorting but that the outcome is influenced by tension such that the two work in concert. Our analysis of the dynamics of sorting suggests differential cadherin expression plays a role in cell sorting in the first 24h and thereafter, the structures become locked. We have determined the efficiency of ETX-embryo formation at different time points. To do this, we collected all structures that had formed at intervals after the time of cell seeding), and analyzed the spatial organization of cell types to quantify well-sorted ETX embryos. The proportion of correctly sorted ETX-embryos plateaued at 15% after just the first day of culture, the sorting phase (Figure R4A). These data suggest that the three cell types can sort only within the first day after seeding, and that subsequent compartment consolidation prevents further sorting. We infer that ETX-embryos undergo two distinct, consecutive phases of morphological changes: sorting followed by consolidation. Furthermore, we hypothesise that cells can no longer sort during the consolidation stage due to the low mobility of cells. To test this hypothesis, we analyzed the mobility of ES, TS and XEN cells by time-lapse imaging as they underwent selforganization into ETX structures (Figure R4B). Each stem cell type expressed a distinct fluorescent marker: ESH2B-CFP, TS-eGFP, XEN-H2B-RFP (Figure R4B), allowing computational analysis of the cell tracks of the individual cell types. We found that all cell types move during the cell sorting stage, whereas cells become relatively immobile during the tissue consolidation stage (Figure R4C and D). Together, those data suggested that after cells sort into the position, they become trapped and cannot sort any more within the time scale we studied. We therefore anticipate that sorting will be completely inhibited in KD condition.

We have added these data in the manuscript as Figure S3A-D.

Figure R4. (A) Time course of formation of correctly-sorted ETX embryos following seeding. 0.5-h: 0/245 structures (n=4); 12-h: 83/1523 structures (n=4); 24-h: 370/2653 structures (n=4) 48-h: 343/3106 (n=4) 72-h: 483/3743 structures (n=4). (B) Live cell imaging and tracking. H2BRFP-XEN (green), H2B-CFP-ES (red) and EGFP-TS (blue) were overlaid with Imaris cell-tracking spheres. (C) Quantification of mobility for different types of cells during self-organization. (D) The bar graph shows the average mobility for different cell types during self-organization at different time ranges after cell seeding. ** $p < 0.01$, *** $p < 0.001$ represents significance, NS indicates no significant difference.

- On a related note, in the cadherin OE cases that improve sorting efficiency, it would be appropriate to cite theory papers showing that larger quantitative differences in cadherin expression between distinct cell types increases the rate of sorting (<https://doi.org/10.1371/journal.pone.0024999>). Again, I wonder whether it is the rate of sorting or the robustness of sorting that is increased in Figure 3H. Performing time course measurements of sorting efficiency may address this question.

Response: To answer this question, we had fixed ETX structures made from E-cadherin-OE-ES, P-cadherin-OETS and XEN cells at different time points, and stained them with lineage markers to quantify cell sorting efficiency, ETX structures made from WT cells were used as controls (Figure R4). We observed that cell sorting rate was increased in cadherin OE structures, as around 30% structures were already well-sorted at 12 hours after cell seeding in structures made from cadherin overexpressed cells, in comparison with 6.8% on wt cadherin structures. This result indicates that the sorting rate is increased upon cadherin OE.

We have added these results as Figure S4E in the revised manuscript.

Figure R4. Examples of structures made from E-cadherin-OE-ES, P-cadherin-OE-TS and XEN cells, stained at different time points to reveal ES cells (Oct4, red), TS cells (Tfap2c, blue) and XEN cells (Gata4, green). Scale bar represents 100 μ m. Quantification shows time course of formation of correctly-sorted ETX embryos following seeding. Control: 12-h: 24/332 structures; 24-h: 83/531 structures; 48-h: 71/448 structures; 72-h: 51/378 structures. Cadherin OE: 12-h: 80/276 structures; 24-h: 139/385 structures; 48-h: 136/374 structures; 72-h: 151/455 structures. N=3 for all conditions. *p<0.05 **p<0.01, ***p<0.001 represents significance between control and Cadherin OE ETX at the same time point.

- The authors mention that the broad range of measured cohesion forces within a population could explain why such a large fraction of the EXT-embryos exhibit mis-sorted configurations. One way to test this hypothesis would be to use bootstrapping of the interaction force measurements to determine what fraction of embryos, whose cells have interaction forces randomly sampled from the measured distribution, are predicted to have the proper hierarchy to produce the correct sorted pattern.

Response: We thank the reviewer for this excellent suggestion. As explained above (Figure R2), we have used the CPM as a generative model to test whether measured distributions of adhesion forces (AFM) are consistent with the distributions of conformations we observe upon sorting in vitro. The CPM is parameterized by an interaction matrix, defining the strength of adhesion between individual pairs of cells, which we fill by bootstrap-sampling measured adhesion forces among the three stem-cell types.

- Cdh3 OE ES cells localizing to the periphery of the TE layer (Figure 1E-F) can be explained by $T-TE-TE < T-ES(OCECdh3)-TS < T-ES(OCECdh3)-ES(OCECdh3)$, according to the differential adhesion hypothesis. It implies the OE of Cdh3 allows strong enough binding to TE cells to overcome the native ES-ES interactions mediated by Cdh1.

Response: We totally agree with this comment and now indicate this in the revised manuscript (page 3)

- In the main text, the figure reference for the contact angle force inference (Figure 2D) is mislabeled as Figure 1D.

Response: We corrected this typo. Thank you.

- Figure 1 PE and VE seem to be used interchangeably. The term VE is used in Figure 1D, but the same layer is labeled PE in Figure 1A. It might be helpful to use a consistent terminology.

R: We use primitive endoderm (PE) for pre-implantation embryo and visceral endoderm (VE) for post-implantation embryo. We have now clarified this in the figure legends. Thank you.

- Figure 2 o You might label the y-axis of G "Homotypic cohesion force" for clarity.

Response: We changed the y-axis of Figure 2G into "homotypic cohesion force" as suggested.

o For Figure 2E, it would be nice to have statistical significance for all comparisons represented.

Response: We now provide statistical significance for all comparisons represented in Figure 2E.

o Will you please discuss in the text why KD of either Cdh1 or CDh3 is sufficient to abrogate homotypic adhesion of TS cells? I would expect that both would have to be removed to eliminate adhesion.

Response: We anticipate that this likely reflects the dosage of cadherins on the membrane, such that downregulation of one cadherin is sufficient to decrease cell-cell adhesion force below a critical threshold. We now discuss it in the text (page 5)

Decision Letter, first revision:

2nd June 2022

Dear Magdalena,

Thank you for submitting your revised manuscript "Stem cell derived synthetic embryos self-assemble by exploiting cadherin codes and cortical tension" (NCB-LE47883A). It has now been seen by the original referee 3 and their comments are below. The reviewer finds that the paper has improved in revision, and therefore we'll be happy in principle to publish it in Nature Cell Biology, pending minor revisions to comply with our editorial and formatting guidelines.

Thank you again for your interest in Nature Cell Biology Please do not hesitate to contact me if you have any questions.

Best wishes,
Stelios

Stylios Lefkopoulos, PhD
He/him/his
Associate Editor
Nature Cell Biology
Springer Nature
Heidelberger Platz 3, 14197 Berlin, Germany

E-mail: stylios.lefkopoulos@springernature.com
Twitter: @s_lefkopoulos

Reviewer #3 (Remarks to the Author):

The authors have fully addressed all my concerns in their revision and really knocked it out of the park with the extension of tension measurements and modeling to fully reconcile their previous experimental observations.

10th June 2022

Dear Dr. Zernicka-Goetz,

Thank you for your patience as we've prepared the guidelines for final submission of your Nature Cell Biology manuscript, "Stem cell derived synthetic embryos self-assemble by exploiting cadherin codes and cortical tension" (NCB-LE47883A). Please carefully follow the step-by-step instructions provided in the attached file, and add a response in each row of the table to indicate the changes that you have made. Please also check and comment on any additional marked-up edits we have proposed within the text. Ensuring that each point is addressed will help to ensure that your revised manuscript can be swiftly handed over to our production team.

We would like to start working on your revised paper, with all of the requested files and forms, as soon as possible (preferably within one week). Please get in contact with us if you anticipate delays.

In recognition of the time and expertise our reviewers provide to Nature Cell Biology's editorial process, we would like to formally acknowledge their contribution to the external peer review of your manuscript entitled "Stem cell derived synthetic embryos self-assemble by exploiting cadherin codes and cortical tension". For those reviewers who give their assent, we will be publishing their names alongside the published article.

Nature Cell Biology offers a Transparent Peer Review option for new original research manuscripts submitted after December 1st, 2019. As part of this initiative, we encourage our authors to support increased transparency into the peer review process by agreeing to have the reviewer comments, author rebuttal letters, and editorial decision letters published as a Supplementary item. When you submit your final files please clearly state in your cover letter whether or not you would like to participate in this initiative. Please note that failure to state your preference will result in delays in accepting your manuscript for publication.

Cover suggestions

As you prepare your final files we encourage you to consider whether you have any images or illustrations that may be appropriate for use on the cover of Nature Cell Biology.

We accept TIFF, JPEG, PNG or PSD file formats (a layered PSD file would be ideal), and the image

should be at least 300ppi resolution (preferably 600-1200 ppi), in CMYK colour mode.

Nature Cell Biology has now transitioned to a unified Rights Collection system which will allow our Author Services team to quickly and easily collect the rights and permissions required to publish your work. Approximately 10 days after your paper is formally accepted, you will receive an email in providing you with a link to complete the grant of rights. If your paper is eligible for Open Access, our Author Services team will also be in touch regarding any additional information that may be required to arrange payment for your article.

Please note that *Nature Cell Biology* is a Transformative Journal (TJ). Authors may publish their research with us through the traditional subscription access route or make their paper immediately open access through payment of an article-processing charge (APC). Authors will not be required to make a final decision about access to their article until it has been accepted. Find out more about Transformative Journals

Please use the following link for uploading these materials:
[REDACTED]

Best regards,

Adam Lipkin
Staff
Nature Cell Biology

On behalf of

Stylianos Lefkopoulos, PhD
He/him/his
Associate Editor
Nature Cell Biology
Springer Nature
Heidelberger Platz 3, 14197 Berlin, Germany

E-mail: stylianos.lefkopoulos@springernature.com
Twitter: @s_lefkopoulos

Reviewer #3:

Remarks to the Author:

The authors have fully addressed all my concerns in their revision and really knocked it out of the park with the extension of tension measurements and modeling to fully reconcile their previous experimental observations.

Final Decision Letter:

Dear Magda,

I am pleased to inform you that your manuscript, "Stem cell-derived synthetic embryos self-assemble by exploiting cadherin codes and cortical tension", has now been accepted for publication in Nature Cell Biology. Congratulations to you and your team!

Once your paper has been scheduled for online publication, the Nature press office will be in touch to

confirm the details. An online order form for reprints of your paper is available at <https://www.nature.com/reprints/author-reprints.html>. All co-authors, authors' institutions and authors' funding agencies can order reprints using the form appropriate to their geographical region.

Please note that *Nature Cell Biology* is a Transformative Journal (TJ). Authors may publish their research with us through the traditional subscription access route or make their paper immediately open access through payment of an article-processing charge (APC). Authors will not be required to make a final decision about access to their article until it has been accepted. Find out more about Transformative Journals

If you have not already done so, we strongly recommend that you upload the step-by-step protocols used in this manuscript to the Protocol Exchange (www.nature.com/protocolexchange), an open online resource established by Nature Protocols that allows researchers to share their detailed experimental know-how. All uploaded protocols are made freely available, assigned DOIs for ease of citation and are fully searchable through nature.com. Protocols and Nature Portfolio journal papers in which they are used can be linked to one another, and this link is clearly and prominently visible in the online versions of both papers. Authors who performed the specific experiments can act as primary authors for the Protocol as they will be best placed to share the methodology details, but the Corresponding Author of the present research paper should be included as one of the authors. By uploading your Protocols to Protocol Exchange, you are enabling researchers to more readily reproduce or adapt the methodology you use, as well as increasing the visibility of your protocols and papers. You can also establish a dedicated page to collect your lab Protocols. Further information can be found at www.nature.com/protocolexchange/about

With kind regards,

Stelios

Stylianos Lefkopoulos, PhD
He/him/his
Associate Editor
Nature Cell Biology
Springer Nature
Heidelberger Platz 3, 14197 Berlin, Germany

E-mail: stylianos.lefkopoulos@springernature.com
Twitter: @s_lefkopoulos

** Visit the Springer Nature Editorial and Publishing website at www.springernature.com/editorial-and-publishing-jobs for more information about our career opportunities. If you have any questions please click here.**